# Research Progress, Hotspots, and Evolution of Nighttime Light Pollution: Analysis Based on WOS Database and Remote Sensing Data

Chenhao Huang [1,2], Yang Ye [1,2,*], Yanhua Jin [3] and Bangli Liang [3]

1    Institute of Spatial Information for City Brain, Hangzhou City University, Hangzhou 310015, China
2    College of Environmental and Resource Sciences, Zhejiang University, Hangzhou 310058, China
3    Zhejiang Territorial Spatial Planning Institute, Hangzhou 310007, China
*    Correspondence: yeyang@zucc.edu.cn; Tel.: +86-153-9714-9370

**Abstract:** With the rapid development of the global economy, the over-expansion of outdoor artificial light at night (ALAN) in cities has led to increasingly severe light pollution worldwide. More and more studies have paid attention to the problem of light pollution, but there is still a lack of systematic literature review on nighttime light pollution in terms of research progress, hotspots, and its evolutions. For this purpose, this study firstly analyzed current research actuality and trends about nighttime light pollution via a comprehensive retrospect of pertinent literature and summarized the adverse effects and monitoring technologies of light pollution by VOSviewer-based keyword co-occurrence technique. Additionally, the study explored the variation tendency of nighttime light pollution in typical countries from 2013 to 2021 based on remote-sensing data and further proposed management suggestions to protect the nighttime environment. The results indicate that the research popularity of nighttime light pollution has been increasing recently, especially after the opening of diversified remote-sensing data in 2012; the main research topics are dominated by adverse effects and monitoring technologies, where the latter is represented by ground survey and remote-sensing observation; the total levels of ALAN intensity are relatively high in most developed countries, but the mean and per capita values are decreasing, and the above phenomenon in developing countries show the opposite trend. This study expects to integrate the literature analysis method and remote-sensing data to demonstrate the research status and variation trends of nighttime light pollution systematically so as to provide scientific references for the assessment and management of the nighttime light environment.

**Keywords:** nighttime light pollution; adverse effects; monitoring technologies; variation tendency; nighttime environment management

## 1. Introduction

Over the past half-century, the intensity of artificial light at night (ALAN) has increased apparently, driven by the rapid innovation of electric lighting technology and transportation infrastructure [1]. The advent of ALAN not only improves nighttime safety for human beings but also brings vitality to the city, and the level of ALAN also reflects the progress of urbanization to some extent [2]. However, along with the continuous expansion of outdoor lighting in recent years, the massive application of new illuminants represented by light-emitting diodes (LEDs) has further increased the brightness of the urban environment, which has already exceeded the actual need [3], resulting in the prominent phenomenon of light pollution around the world.

In urbanization, people have lit up almost all artificial environments, including roads, bridges, shopping malls, airports, parking lots, green belts, residential buildings, and industrial parks. The aggregation of high-intensity ALAN illuminates the target area as well as its surroundings [4–6], affecting zones up to one hundred kilometers away from

the city edge [7]. Surveys show that in 2016, more than 80% of the world's population lived under skies with varying degrees of light pollution; more than 30% had difficulty seeing the Milky Way directly [8,9], while the area with ALAN grew at a rate of 2.2% per year [4,10]. More importantly, recent research suggests that the emissions of ALAN have increased by at least 49% between 1992 and 2017 [11–13]. Considering the difficulties of existing satellite sensors to detect radiance information from new light sources such as LEDs, the actual increase in global ALAN during this period could be as high as 270%, with gains of up to 400% in several regions [8,14]. Thus, excessive expansion of ALAN under the background of rapid urbanization has become an established status, while a deeper understanding of the specific supply and demand of ALAN is urgent. Existing studies have already found that an oversupply of ALAN is prevalent in most cities worldwide [14–17]; for instance, Ye et al. analyzed the supply and demand of ALAN in Hangzhou, China, from the citizens' perspective, indicating a significant mismatch between local ALAN provision and the actual needs [18].

Uncontrolled utilization of ALAN changes the rhythm of natural light, aggravating the light spillover from the city center to the suburbs hundreds of miles away and causing nonnegligible issues [19]. Numerous reliable studies have shown that light pollution hinders astronomical observations, causes ecological deterioration, endangers human health, and contributes to the waste of energy and even the loss of the cultural value of the dark night. Concretely, excess ALAN strongly interferes with the light from celestial bodies, making it problematic for astronomical instruments to seize them accurately [20]. Long-term exposure to inappropriate ALAN will induce diversiform physical and mental diseases and even endanger personal safety [2,21–27]. Not only humans but also other organisms on Earth suffer from ALAN, as evidenced by the decay of biodiversity and the disruption of life rhythms [28–42]. Behind light pollution is often a disproportionate consumption of electricity compared to logical demand, which is inevitably accompanied by energy waste and excessive carbon emissions [43–45]. Worse still, the natural ornamental value and priceless cultural connotations carried by the night sky are also gradually passing away [46,47].

For now, urban sprawl is still taking place globally, and this tendency may not be reversed in the coming decades [48], which implies that the unfavorable effects of nighttime light pollution will become more and more severe without the support of precise monitoring techniques and suited management strategies. Therefore, it is imperative to perform a literature review of nighttime light pollution to deeply understand the status and provide meaningful references to optimize urban lighting regulation. To achieve this goal, this study was conducted in the following four parts. First of all, the data and methods required for literature analysis and ALAN trend analysis were introduced, respectively. Second, based on the results of the literature search and screening, the research progress in the field of nighttime light pollution, including key directions, quantity of annual publications, and major publishing countries, was presented. Third, based on text mining and keyword co-occurrence analysis, research hotspots and significant advances were extracted and reviewed, including the adverse effects of nighttime light pollution and mainstream monitoring techniques. Last but not least, satellite data were adopted to reveal the evolution of nighttime light pollution in typical countries (G20) over a long time series (2013–2021), and feasible suggestions for lighting regulation were proposed accordingly.

This paper is organized as follows. Section 1 describes the background of the study. Section 2 describes the data sources and research methods applied. Section 3 presents the result of the statistical analysis of the literature, followed by summarizing the research hotspots and investigating the evolutionary trends of nighttime lighting. Section 4 carries out the discussion part. Section 5 lists the main conclusions.

## 2. Materials and Methods

### 2.1. Materials

2.1.1. Literature Database

The importance of bibliographic databases (DBs) as the primary source of publication metadata and citation metrics in scientific research cannot be overemphasized. Compared to other highly specialized or emerging bibliographic databases, such as ResearchGate, PubMed, Microsoft Academic, and Open Citations, Web of Science (WOS) and Scopus have evident strengths in aspects of literature coverage, information completeness, and convenience of the interface [49,50]. Both have become prominent bibliographic databases on the Internet today and are recognized as two of the most comprehensive literature data sources [51]. WOS is the first international bibliographic database with extensive coverage and has over time become the most influential bibliographic source for journal selection, research assessment, bibliometric evaluation, and other scientific tasks. In addition, Scopus, launched by Elsevier in 2004, has proven its reliability in recent years, earning an equal status as a multidisciplinary bibliographic database. Many studies have demonstrated that the literature collections covered by WOS and Scopus are highly overlapping, but the degree of overlap varies considerably across disciplines [52]. For example, large-scale journal-level comparisons showed that WOS had a higher proportion of literature in the natural sciences and engineering, while Scopus was more biased toward biology and medicine [53]. A similar picture of the disciplinary distribution was also presented in surveys conducted on publishing institutions or application areas [49].

WOS is an integrated academic information platform developed and maintained by Thomson Reuters that includes more than 18,000 authoritative and high-impact academic journals from across the world, covering fields including but not limited to natural sciences, engineering, biomedical sciences, social sciences, arts, and humanities [54]. WOS includes most of the relevant papers written by top scholars and thus has a high impact and wide international scope. To this day, it has become an internationally recognized database reflecting the standard of scientific research and is currently the most preferred source of citation data [55]. Therefore, considering that natural sciences are more dominant in the discipline of nighttime light pollution research, and to ensure the reliability and comprehensiveness of the literature analysis, the WOS database was adopted as the main source of literature information in this study.

2.1.2. Remote Sensing Data

Nighttime light imagery observed by satellites can indicate the magnitude of Earth's lighting brightness on a large scale and thus has been widely used in estimating light pollution these days [56]. Launched in 2011, the Suomi-National Polar-orbiting Partnership (S-NPP) satellite with the Visible Infrared Imaging Radiometer Suite (VIIRS) was designed to monitor the atmospheric environment initially [57]. The Day/Night Band (DNB) of NPP-VIIRS is competent for providing cloud-free daily nighttime light data for ALAN studies [58]. A prominent feature of the DNB data is their effective detection of electric lighting signals within the range of human activity. This ultra-sensitivity in low-light conditions enables the VIIRS DNB products to better monitor both the magnitude and signature of nighttime phenomena and anthropogenic sources of light emissions [59]. On the other hand, due to this high sensitivity, some reflected light from snow and sand will also be captured by the S-NPP satellite, thus creating background noise and anomalies in the images [60]. Since the launch of the satellite, VIIRS data have been processed, produced, and released by the National Oceanic and Atmospheric Administration (NOAA) and the National Aeronautics and Space Administration (NASA). A series of preprocessing steps have been applied to exclude partial noise, cloud cover contamination, solar and lunar interference, and features such as fires, volcanoes, and flares that are not relevant to ALAN, giving rise to data products with different resolutions and qualities, such as yearly integrated product, monthly integrated product, and daily black marble product [61].

Therefore, in this study, the open-source NPP-VIIRS nighttime light data were downloaded from the Light Pollution Map website (www.lightpollutionmap.info) (accessed on 12 April 2023) to evaluate the nighttime light pollution of G20 countries from 2013 to 2021. The NPP-VIIRS product displayed on the website, codenamed VNP46A4, known as VIIRS/NPP Lunar BRDF-Adjusted Nighttime Lights Yearly L3 Global 15 arc second Linear Lat Lon Grid, is the fourth nighttime lights product in the black marble suite, which provides annual composites generated from daily atmospherically and lunar-BRDF-corrected NTL radiance since 2012 to eliminate the effects of extraneous artifacts and biases. VNP46A4 contains 28 layers, which provides information on the NTL composite, the number of observations, quality, and standard deviation for multi-view zenith angle categories (near-nadir, off-nadir, and all angles), their snow-covered and snow-free statuses besides land-water mask, and latitude and longitude coordinate information [62]. The cloud-free products generated by NASA's Black Marble algorithm have been adjusted for various factors such as atmospheric, terrain, lunar BRDF, thermal, and straylight effects. This correction for nighttime radiance allows for the more accurate retrieval of nighttime lights over short time periods while reducing background noise, which enables quantitative detection and analysis of daily, seasonal, and annual variations. To remove residual background noise, the NTL composite values for radiation brightness less than $0.5 \text{ nW·cm}^{-2}\text{·sr}^{-1}$ were set to zero. Moreover, countries close to the Arctic Circle (e.g., United Kingdom, Russia, and Canada) have most of their regions at latitudes with high auroral activity, which causes NTL data to differ significantly from the actual light emission values in these regions. As a result, these aurora-contaminated pixels were filled with gap-filling values [63]. Although the uncertainty in VNP46A4 had not been completely eliminated after the data preprocessing described above, such as the residuary background noise and insensitivity to near-blue wavelengths, the level of impact is slight enough to be competent for subsequent analysis based on numerical statistics.

### 2.2. Methods

#### 2.2.1. Database Search Strategies

We conducted an extensive search of all sub-databases in the WOS database in "advanced search" mode and collected subject-related articles published in various languages. The exact search field was set as "Topic". The search formula was: TS = (("night" OR "nighttime" OR "night-time") AND "light pollution"* AND ("ALAN" OR "artificial light at night"* OR "NTL" OR "nighttime light"* OR "satellite" OR "remote sensing"*)), where several Boolean operators can be applied to assist the search. "AND" was utilized to connect parallel keywords, "OR" was used to connect branching keywords, the " " double quotes denoted that the phrase must be searched in its entirety rather than split up, and the "*" suffix indicates a wildcard. In addition, the search date range was set from 1 January 2000 to 31 December 2022 to fully retrieve documents published during the first 23 years into the 21st century. In addition, the WOS query link is: https://www.webofscience.com/wos/alldb/summary/77160b0a-f686-41e7-a93c-06ae193f6066-7ecc4c6b/relevance/1 (accessed on 12 April 2023). In addition, considering factors such as completeness, availability, and representativeness, essential rules were developed for inclusion and exclusion to ensure the documents obtained qualified enough for subsequent analysis when skimming preliminary search results. Fundamental information of the collected literature is specified in Table 1.

**Table 1.** Summary of the literature derived from the WOS database search.

| Items | Explanation |
|---|---|
| Used ub-databases | All sub-databases contained in WOS, which includes Web of Science Core Collection (with SCIE, SSCI, A&HCI, and ESCI), BIOSIS Citation Index (BCI), Chinese Science Citation Database (CSCD), Derwent Innovations Index (DII), the food science resource (FSTA), Korean Journal Database (KCI), MEDLINE, SciELO Citation Index, and Zoological Record |
| Search keywords | "night", "nighttime", "night-time", "light pollution", "ALAN", "artificial light at night", "NTL", "nighttime light", "satellite", "remote sensing" |
| Time span | 1 January 2000–31 December 2022 |
| Search field tag | Topic (including Title, Abstract, Author Keywords [1], and Keywords Plus [2]) |
| Search formula | TS = (("night" OR "nighttime" OR "night-time") AND "light pollution"* AND ("ALAN" OR "artificial light at night"* OR "NTL" OR "nighttime light"* OR "satellite" OR "remote sensing"*)) |
| Inclusion criteria | The type of literature should be constrained to articles, review articles, meeting papers, and books, and the theme should focus on aspects related to nighttime light pollution |
| Exclusion criteria | Preprints and documents that are not full text, not available online, and not related to the topics already specified above will be excluded |

[1] One of the two types of keywords in WOS, i.e., keywords provided by the original author when submitting the document. [2] The other type of keywords in WOS, extracted from the titles of the literature and its references by automatic clustering algorithms.

### 2.2.2. Literature Selection Methods

The selection steps of the retrieved literature were divided into four steps, and the specific screening process is shown in Figure 1.

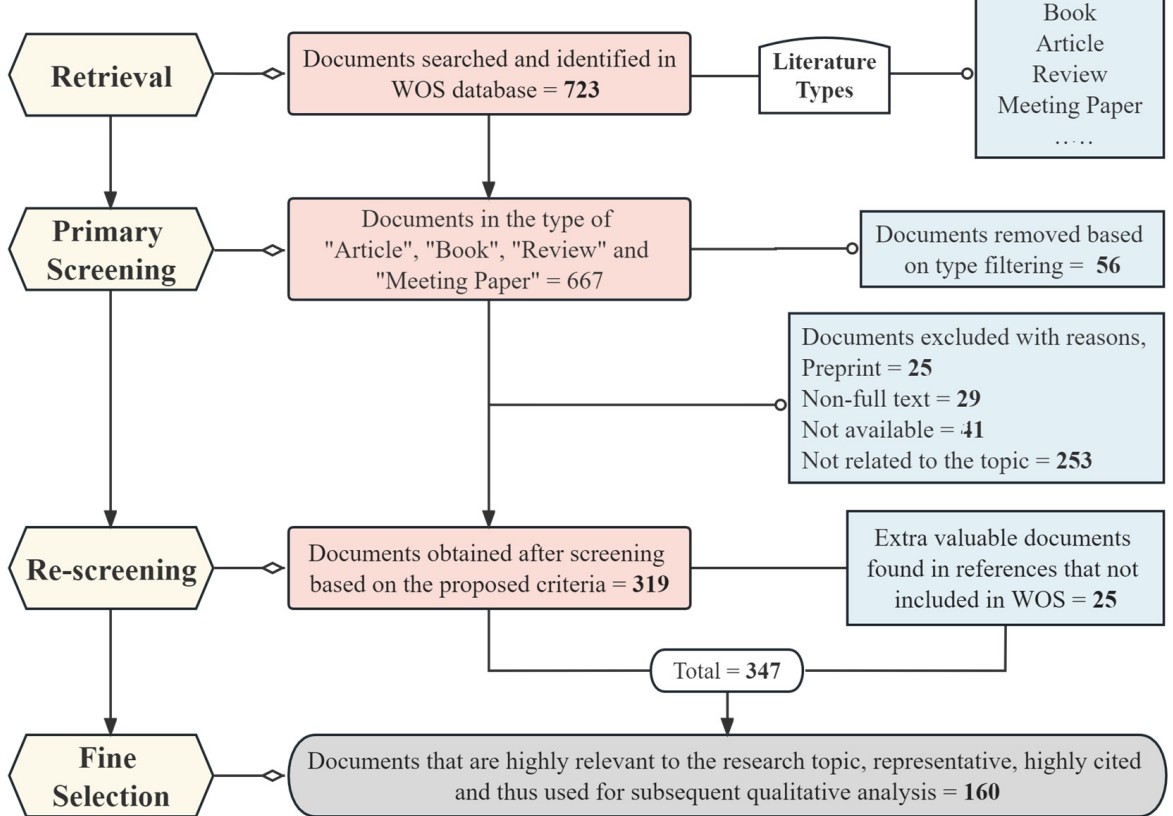

**Figure 1.** The process of literature selection.

1.  **Retrieval.** A total of 723 unduplicated documents were retrieved from the Web of Science database, including a variety of types such as books, articles, reviews, patents, abstracts, meeting papers, editorial material, and data papers.
2.  **Primary screening.** For the 723 documents retrieved, the type of articles was restricted to "Book", "Article", "Review", and "Meeting Paper" according to the established exclusion criteria in Table 1, and a total of 667 documents were obtained after refinement.
3.  **Re-screening.** The 667 documents obtained from the initial screening were skimmed, focusing on the keywords and abstracts of the articles, and those that were not relevant to the research topic, not available, and without full text (including unpublished preprints) were screened and excluded. In this way, 319 documents meeting the requirements were acquired after screening. Furthermore, another 25 thesis or review papers not included in WOS were selected as additional data when browsing the references. Finally, a total of 344 documents were obtained.
4.  **Fine selection.** After a thorough reading of the 344 documents, high-quality representative documents that are highly compatible with the research topic and in the top 160 cited were selected as the key reference contents for subsequent analysis (see Supplementary Materials for a complete list of literature).

2.2.3. Analysis of Keyword Co-Occurrence

A co-occurrence analysis map of keywords based on the primary screening results was constructed with the help of VOSviewer, a tool for building visual bibliometric networks developed by the Science and Technology Research Center of Leiden University in the Netherlands [64].

In principle, VOSviewer constructs maps based on co-occurrence matrices. The construction of the analysis map is a three-step process. Firstly, a similarity matrix is calculated based on the co-occurrence matrix. Secondly, a preliminary map is constructed by applying the VOS mapping technique to the similarity matrix. Finally, based on predefined parameters, the map is translated, rotated, and reflected [65–67]. More specifically, the similarity matrix in step 1 is obtained based on the keyword co-occurrence matrix by calculating the association index with the aim of correcting for differences in the total number of occurrences or co-occurrence of items. The VOS mapping technique in step 2 is the pivotal operation for constructing the network map, whose idea is to minimize the weighted sum of the squared Euclidean distances between all item pairs, and the higher the similarity between two items, the higher the weight of their squared distances in the sum [68]. Step 3 is performed to make the results more stable and reliable by a series of transformations to ensure that the same global optimal map is generated from the same co-occurrence matrix. Further details about VOSviewer and its technical implementation can be found in the instruction manuals and related publications available on the official website (https://www.vosviewer.com/) (accessed on 12 April 2023).

VOSviewer also provides a window that displays the map in 3 ways, such as Network Visualization, Overlay Visualization, and Density Visualization, each emphasizing a different aspect. It also offers zoom, scrolling, and search functions to facilitate detailed examination of the map, which is particularly useful for analysis involving substantial entries [69]. In contrast, most existing computer software for bibliographic mapping are not yet capable of displaying maps in such a satisfactory manner.

Overall, the main advantages of VOSviewer over other bibliometric software are its efficient text analysis capabilities and its powerful graphical display functions, making it ideal for processing and visualizing massive data. Here, the text mining and visualization functions provided by VOSviewer were adopted for generating co-occurrence networks of important terms extracted from the scientific literature.

There are two types of keywords recorded per document in WOS: author keywords and keywords plus. Author keywords are keywords provided by the original author at the time of submission. Keywords plus are correlative keywords extracted from the titles

of documents and their cited references, usually generated automatically by clustering algorithms, and not necessarily located in the title of the article or appearing as author keywords [70]. Many studies have shown that keywords plus are able to capture the content of the literature with greater depth and diversity than author keywords and are often employed as a key metric in the bibliometric analysis [71–73]. Keywords plus have also been widely applied to identify research trends and hotspots in many domains, such as climate change [74], air pollution [75], biomedicine [76], etc. Therefore, to ensure the objectivity of keyword co-occurrence analysis, this study adopted keywords plus as the basic units of analysis on VOSviewer.

Since a considerable portion of the keywords extracted from the topic of the literature is usually composed of two or more separate words and synonyms among them are inevitable, this study clustered the words or phrases with the same meaning by importing a pre-created thesaurus file.

In order to remove irrelevant miscellaneous items and make the visualized network map more focused on the topic, the list of keywords after synonym merging was reviewed again, and words with no real meaning and completely irrelevant to the study topic were eliminated. After that, for the sake of aesthetics and clarity, this study adopted a threshold approach to filter out some of the keywords that occupy a relatively small percentage. That is, keywords that occurred less frequently than a certain value would be removed from the map. This specific value of the threshold was decided after numerous tests for the purpose of constructing the most intuitive keyword co-occurrence map. In this study, the threshold was set at 10.

### 2.2.4. Analysis of Nighttime Light Pollution in Typical Countries

The Group of Twenty (G20) was taken as a case study (except the European Union). The G20 is recognized as the main forum for international economic cooperation, with membership from six continents, balancing the interests of developed and developing countries. According to statistics, the G20 has 2/3 of the world's population, 55% of the world's land area, and 86% of the world's GDP (gross domestic product), and the intensity of nighttime activities is statistically the highest in the world, so it is well represented for research [77].

Here, to accurately quantify the nighttime light environment, three statistical indicators were proposed: total nighttime light (noted as ***Rad. Sum***), nighttime light per unit area (noted as ***Rad. Mean***), and nighttime light per 1000 people (noted as ***Rad./1k Pop***). Each indicator is explained specifically as follows.

- ***Rad. Sum*** was defined as the sum of the radiance values of each pixel cell in the study area, which represented the overall ALAN intensity and was also associated with the total economic volume of the country to some extent;
- ***Rad. Mean*** was set as the ratio of the total nighttime light (***Rad. Sum***) to the country's territorial area, reflecting the illumination per unit area and indirectly revealing the density of nighttime economic activity in the country;
- ***Rad./1k Pop*** was expressed as the average ALAN per 1000 people in the target area, i.e., the ratio of total nighttime light (***Rad. Sum***) to the number of inhabitants (in thousands) in the country, which was an indicator that characterized the intensity of light radiation per capita and showed the degree of impact of ALAN on people's lives within the territory.

After the indicator system was established, a line chart was drawn, and linear fit parameters were shown to describe the evolution of nighttime light pollution in each G20 country over the long term. To further characterize the year-by-year variations of the three indicators in each G20 country, this study adopted the Mann–Kendall trend test to explore whether they had a monotonic variation trend.

The Mann–Kendall test (MK test) is a non-parametric trend test for time series data originally proposed by Mann in 1945 and further refined by Kendall [78]. This method is convenient to operate and does not require the samples to follow a particular distribution

(e.g., normal distribution), nor is it disturbed by a few missing values or anomalies [79,80]. In the MK test, the null hypothesis ($H_0$) is that there is no monotonic trend in the sample, and the alternative hypothesis ($H_1$) is that there is a monotonic trend in the sample. For mathematical expression, $H_0$ assumes that the time series data ($X_1, X_2, \ldots, X_n$) are independent, random variables with the same sample distribution; $H_1$ assumes that for all i, j $\le$ n (i $\ne$ j), the distributions of $X_i$ and $X_j$ are identical. The statistic S of the test is calculated as follows:

$$S = \sum_{i=1}^{n-1} \sum_{j=i+1}^{n} \mathbf{sgn}(X_i - X_j) \tag{1}$$

where $X_i$ and $X_j$ are the observed values corresponding to the i-th and j-th time series, respectively, and i < j. $\mathbf{sgn()}$ is the sign function and is calculated as follows:

$$\mathbf{sgn}(X_i - X_j) = \begin{cases} 1 & (X_i - X_j) > 0 \\ 0 & (X_i - X_j) = 0 \\ -1 & (X_i - X_j) < 0 \end{cases} \tag{2}$$

When n $\ge$ 8, the statistic S roughly follows a normal distribution with a mean value of 0 (i.e., $\mathbf{E}(S) = 0$) without considering the existence of equivalent data points in the series. The variance can be calculated by the following equation:

$$V_{ar}(S) = \frac{n(n-1)(2n+5)}{18} \tag{3}$$

Further, the standardized test statistic Z is calculated as follows:

$$Z = \begin{cases} \frac{S-1}{\sqrt{V_{ar}(S)}} & S > 0 \\ 0 & S = 0 \\ \frac{S+1}{\sqrt{V_{ar}(S)}} & S < 0 \end{cases} \tag{4}$$

In the bilateral trend test, for a given confidence level (significance level) $\alpha$, if $|Z| \ge Z_{1-\alpha/2}$, the original hypothesis $H_0$ is unacceptable, i.e., there is a significant upward or downward trend in the time series data at the confidence level of $\alpha$. A positive value of Z indicates an upward trend, and a negative value indicates a downward trend. Empirically, absolute values of Z greater than or equal to 1.645, 1.96, and 2.576 indicate that the significance tests with 90%, 95%, and 99% confidence levels are passed, respectively [81].

In addition, the magnitude of the trend in the time series data can be measured by the slope $\beta$, which can be calculated as follows:

$$\beta = \mathbf{median}\left(\frac{X_j - X_j}{j - i}\right) \ \forall 1 < i < j < n \tag{5}$$

where $\mathbf{median()}$ is the operation of taking the median value. A positive value of $\beta$ indicates an "increasing trend", while a negative value of $\beta$ indicates a "decreasing trend". The larger the absolute value of $\beta$, the more pronounced the tendency is.

## 3. Results

### 3.1. Statistical Analysis of Literature

After a primary screening of the documents according to the criteria in Section 2.2.1, the 723 obtained papers were statistically analyzed and visualized. Figures 2–4 show the research direction tree statistics, year-by-year trend graph of publication number and citation frequency, and publication country bar statistics, respectively.

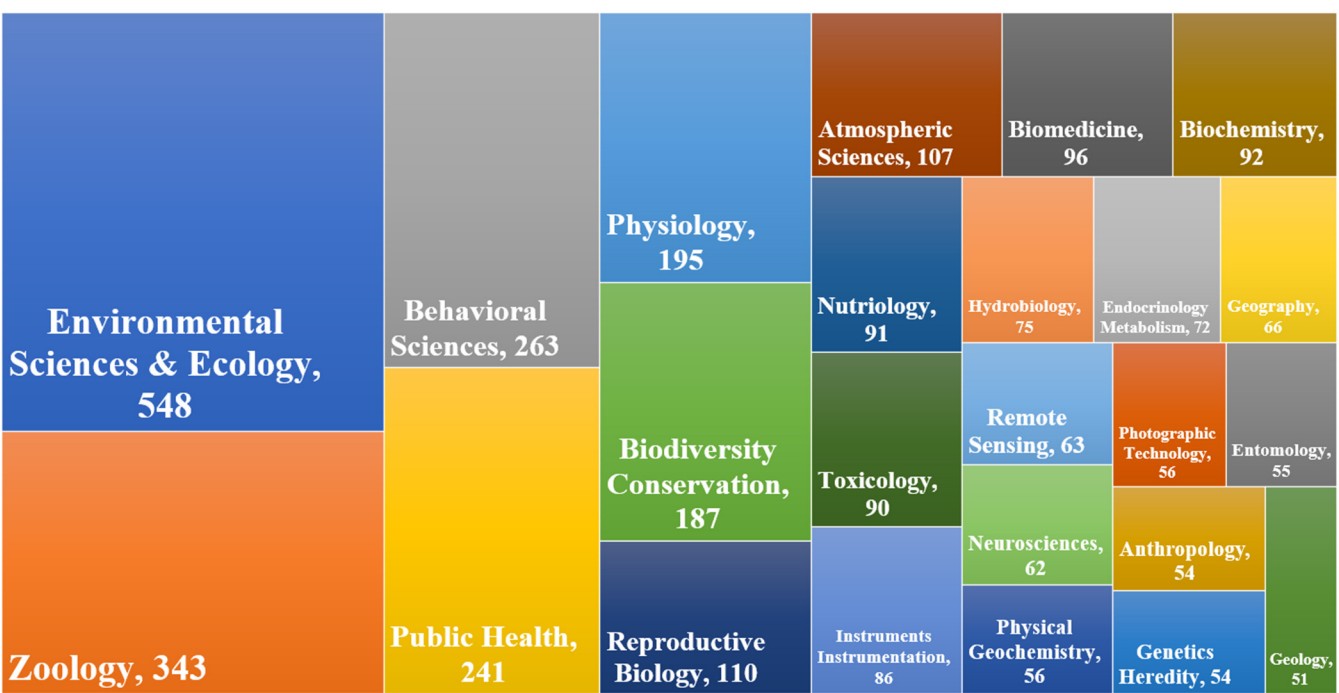

**Figure 2.** The statistical tree chart of research direction. The relative area of each bar in the graph is strictly positively correlated with the number of articles published in that discipline or direction.

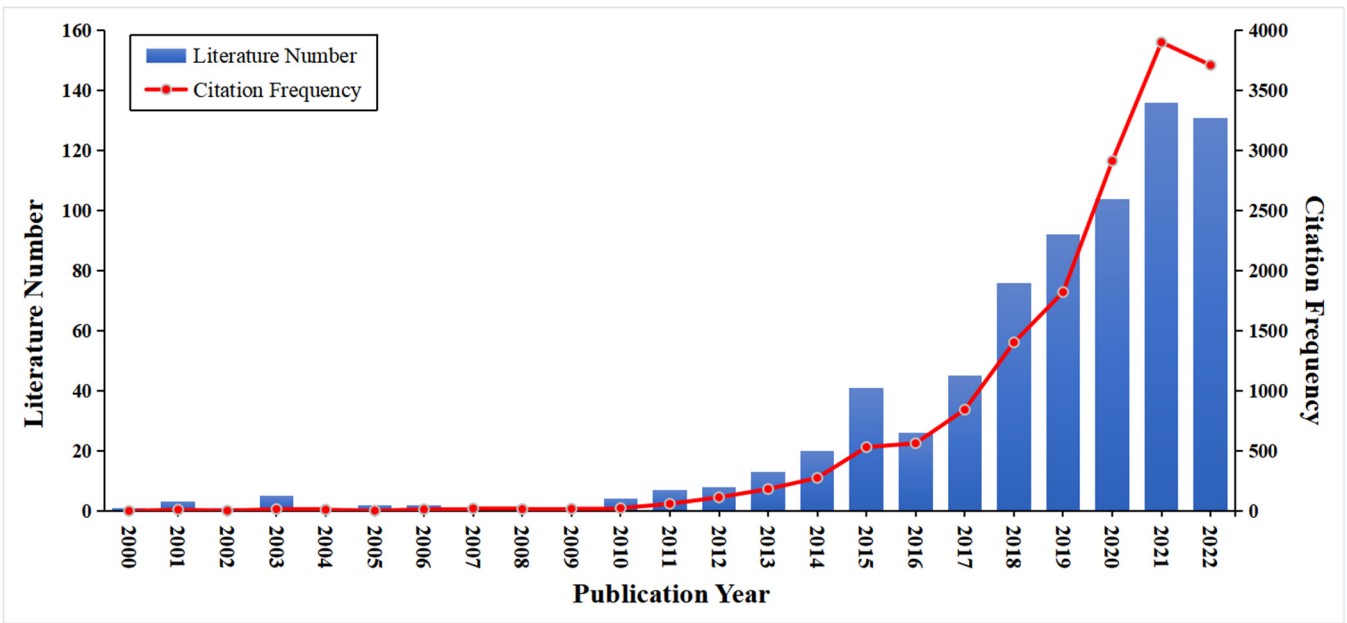

**Figure 3.** Trend graph of number of publications and frequency of citations from 2000 to 2022.

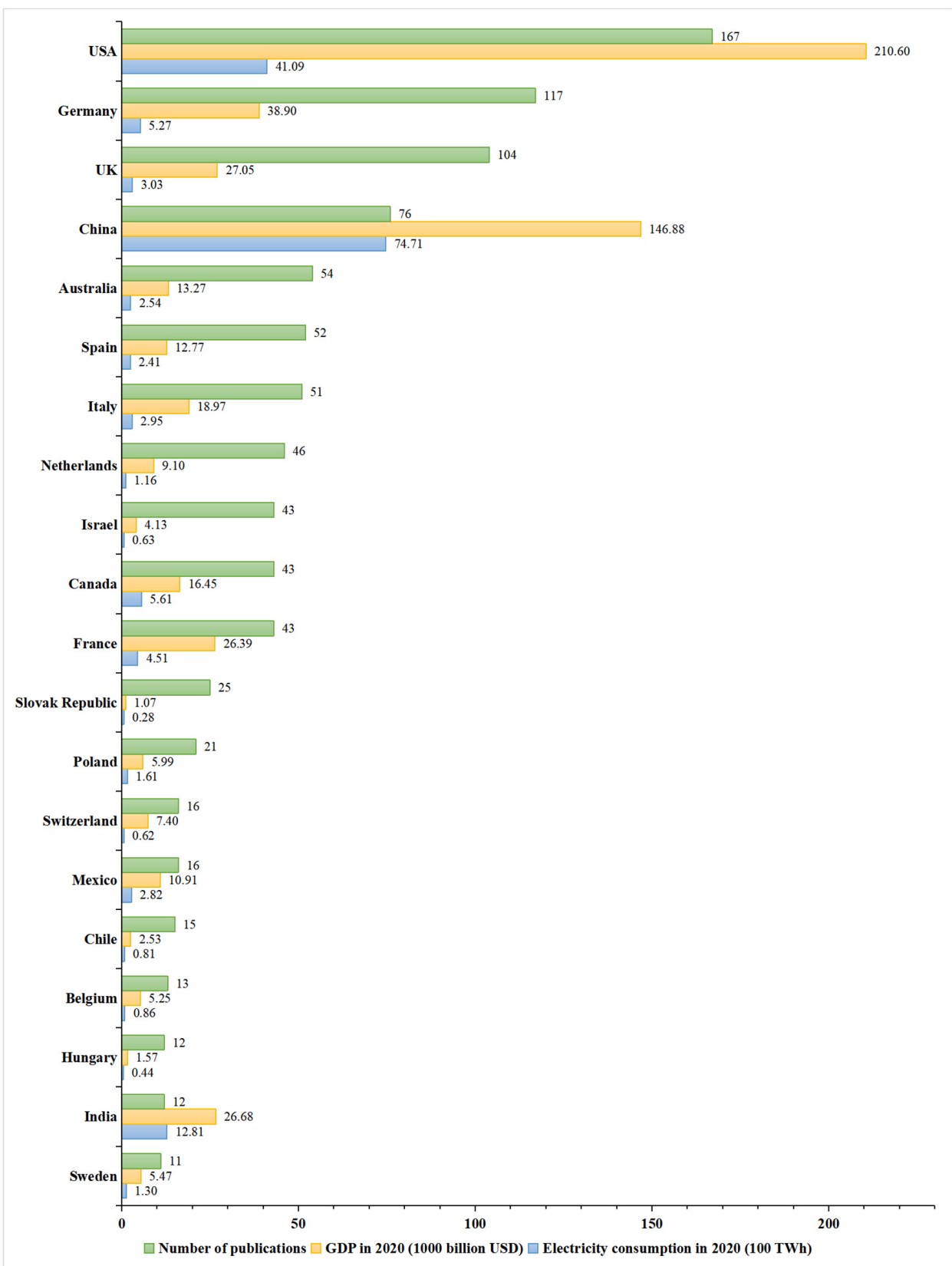

**Figure 4.** Statistical bar chart of the top 20 countries in terms of number of publications and their electricity consumption in 2020.

### 3.1.1. Main Research Fields

The research directions of all 723 documents obtained from the search step are shown in Figure 2 (since each document generally corresponded to multiple research directions, the total frequency count was far more than 723. In addition, to ensure readability, the figure only showed items with a frequency greater than 50. A complete table of research direction counts can be found in Table S1 in Supplementary Materials). As can be seen, during 2000–2022, in the field of nighttime light pollution, the direction with the greatest proportion of scientific research is environmental science and ecology (548 records, accounting for 75.79% of the total), followed by zoology, behavioral science, public health, physiology, biodiversity conservation, reproductive biology, atmospheric science, and so on (in descending order of frequency). It is also worth noting that remote sensing, the main methodological discipline of quantifying nighttime light pollution, also occupied a relatively large share of 8.71%.

### 3.1.2. Trend of Annual Publication Number

As shown in Figure 3, the overall trend showed that the number of publications related to nighttime light pollution during 2000–2022 was generally on the rise. In the first decade of the 21st century, quite a few relevant pieces of literature were published, with a maximum of five publications per year, which indicated that the problem of nighttime light pollution had not received much attention. After 2012, the number of publications increased dramatically, and the frequency of citations also showed a consistent trend. This sudden change may be due to the official opening of the NPP-VIIRS data in April 2012. The availability of NPP-VIIRS data remarkably overcame the defects of its previous generation of nighttime light product, i.e., DMSP-OLS, in aspects of radiation resolution, spatial resolution, on-board calibration, minimum detecting threshold, etc., marking the beginning of a new era of data collection and application [82,83]. Up to now (2020s), urban nighttime light pollution has become a hotspot with broad research prospects.

### 3.1.3. Countries with the Most Publications

Figure 4 illustrates the countries with the highest number of published papers. Considering the typographical constraints, only the top 20 countries in terms of the number of published papers and their GDP and electricity consumption were shown in the figure. The complete table of statistics can be found in Supplementary Materials (Table S2). As can be seen in Figure 4, the five countries with the highest number of publications are the United States, Germany, the United Kingdom, China, and Australia (in descending order), accounting for 48.32% of all countries and nearly half of the publications on related topics. Correspondingly, most of these countries are statistically relatively high in electricity consumption and GDP [53,54] and are important engines of the global economy [84,85]. Hence, an overall positive correlation between the variables can be observed, i.e., the number of publications is essentially synchronized with electricity consumption and GDP in most countries. The above conclusions also confirm that these major global economies are suffering from severe light pollution, and all of them have a much more urgent need for reasonable control of nighttime light pollution, which has driven a boom in academic research in the field of nighttime light pollution.

In addition, from Figure 4, it can be found that the number of publications in the field of nighttime light pollution in a country is not always strictly related to the intensity of economic activity or energy consumption, but in fact is also closely related to a variety of factors such as the economic base, natural resources, policy awareness, and development planning. Among the top 20 countries in terms of the number of papers, developed countries, represented by European countries, have a considerable proportion (70%), and these countries have generally started to pay attention to light pollution problems and carry out explicit practices earlier, and have now accumulated a deeper foundation in legal norms, environmental standards, and management approaches [86,87]. This is probably supported by large research funds, so the enthusiasm of relevant research is accordingly

high. Meanwhile, the International Dark Sky Association (IDA), a professional body dedicated to fighting light pollution, has now certified 21 International Dark Sky Reserves around the world to specifically protect public or private lands with outstanding star quality and nighttime environments and the scientific, natural, educational, cultural, and recreational values they contain [46]. Most of the International Dark Sky Reserves are located in developed countries in Europe and North America, and this special demand for protection has led to a greater commitment to nighttime light pollution research in these countries. On the contrary, developing countries such as China and India, where economic growth is the first priority, are still in the process of changing the concept of balancing economic growth and environmental protection; therefore, the understanding of nighttime light pollution is not comprehensive and profound, and relevant research funding is still insufficient [88]. More detailed discussion of the differences between countries and the experience of European countries in light pollution control can be found in Sections 3.3 and 4.3.

### 3.2. Summary of Research Hotspots

Following the methodology developed in Section 2.2.3, a keyword co-occurrence map was created in VOSviewer based on bibliographic data (i.e., tab-delimited txt format files containing complete records from WOS) (Figure 5). For the visualization, the total link strength was chosen as the quantitative weight for the size of the entry and removed sundry lines with a link strength of less than 2 to enhance visibility, while the rest of the parameters were default. It should be noted that VOSviewer presents the measurement network in three dimensions, so very few terms (e.g., "biodiversity" and "pattern") are not presented integrally because they were obscured by other terms. In this context, the complete list of terms, clusters, and weights is presented as tabulated data in Supplementary Materials (Table S3).

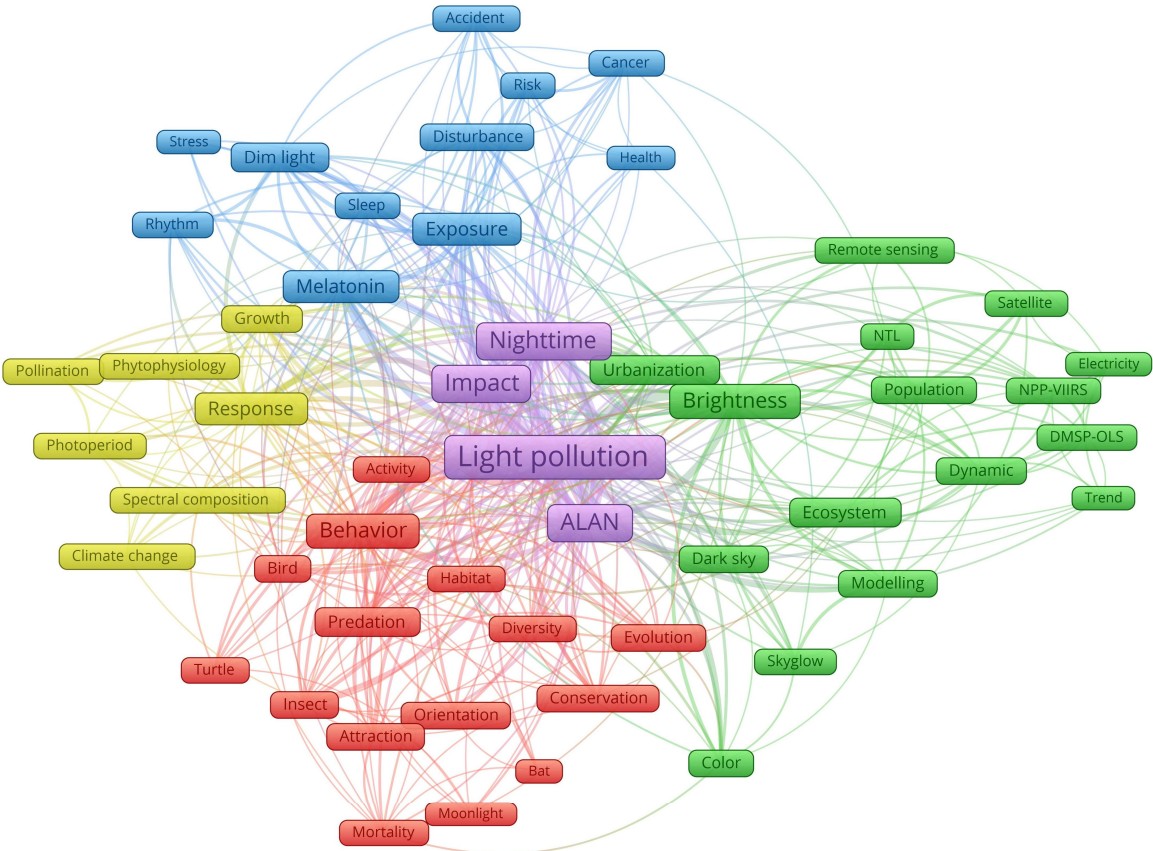

**Figure 5.** Map of keyword co-occurrence analysis (based on VOSviewer).

In Figure 5, the size of an entry is positively correlated with the frequency of occurrence of the term in the literature; entries with the same color indicate that the research directions represented by these terms are relatively close; the thickness of the lines between the entries is positively correlated with the frequency of simultaneous occurrence of the two terms, expressing the relevance of the literature. The closer to the center of the map, the more important the term is and the more relevant it is to the research topic. As can be seen, the 55 eligible keywords were grouped into five categories. These keywords reflected the objects, directions, data, techniques, or concepts that have received great attention in the field of nighttime light pollution. The cluster marked in purple in the middle of the map represented the core concepts of nighttime light pollution research, which were the largest in size with the thickest and most connected lines. The remaining four clusters around the purple cluster could be further divided into two groups according to the research content, i.e., adverse effects (marked in blue, yellow, and red) and research methods (marked in green).

In terms of adverse effects, research hotspots on the effects of light pollution on humans (blue cluster), plants (yellow cluster), and animals (red cluster) were shown separately. Specifically, for humans, studies on the negative impacts of nighttime light pollution included health, sleep, mood, and accidents. As for plants, most studies focused on photoperiod, pollination, and growth under the influence of light pollution. Regarding animals, the research on the damage of nighttime light pollution revolved around the nocturnal behavior, habitat, and evolution of typical light-sensitive animals (birds, turtles, bats, etc.).

In terms of research methods, the terms in green clusters could be broadly divided into monitoring objects, monitoring techniques, and basic data. For the monitoring object, it included the dynamic measurement of ALAN arising in the urbanization and the exploration of the mechanism of the impact of nighttime light pollution on urban and surrounding ecosystems. For monitoring techniques, mainstream light pollution modeling and remote-sensing observations were displayed. For the basic data, widely used core data (e.g., "DMSP-OLS" and "NPP-VIIRS") and auxiliary data (e.g., "population" and "electricity consumption") were exhibited. These conclusions drawn above were also in high agreement with the distribution characteristics of the research directions or disciplines exhibited in Figure 2. It can be found that the direction or discipline entries with a large proportion of area in Figure 2 coincided with the key terms in Figure 5 to some extent.

As mentioned above, two major research hotspots can be summarized as adverse effects and monitoring techniques. Then, the relevant contents in the screened literature were thoroughly read, and a comprehensive review was presented in the following Sections 3.2.1 and 3.2.2.

3.2.1. Adverse Effects of Nighttime Light Pollution

- **Health threat.** The brightness of the environment is one of the most essential factors that synchronizes human circadian rhythms with the Earth's light and dark cycles. A cross-border survey shows that two-thirds of the public believe that light pollution harms their health, and 84% think it affects their sleep quality at night [2,89,90]. Studies also have shown that the threshold of lighting intensity affecting human physiology is so low that only one lux may disrupt circadian rhythms [21,91,92]. Physiologically, when exposed to excessive blue light from outdoor lighting or indoor displays, the eyes may suffer from accidental retinal damage due to oxidative stress [22] and may even lead to reduced levels of melatonin, leading to daily rhythm phase shifts, which increases the incidence of some diseases, including but not limited to vision loss, metabolic disorders, diabetes, obesity, coronary heart disease, tumors, and even cancer [23,93–100]. Psychologically, the altered circadian rhythms triggered by the light environment may lead to alertness, rapid heart rate, sleep disturbances, mood disorders, and other derived problems such as depression, irritability, fatigue, and anxiety [25,26,101]. Newborns exposed to improper light often suffer from sleep

and nutritional issues, possibly stimulating precocious puberty [15,102]. In addition, blinding lighting, such as dazzling glare from intense light sources, is one of the inducements resulting in traffic accidents. Excessively bright lighting will impair the vision of drivers or pedestrians, easily leading to visual disability and making the risk of traffic accidents much higher [27]. In conclusion, the threats of light pollution to the human body are multidimensional, and the resultant excess mortality is currently challenging to gauge accurately.

- **Ecological damage.** Nighttime light pollution affects the feeding, mating, migration, communication, competition, and predation habits of animals, including insects, birds, reptiles, amphibians, and even fish. For example, light pollution interferes with the navigational abilities of nocturnal insects such as moths, many of which congregate around light sources until exhaustion [30,103–106]. Nocturnal flowering plants that depend on these insects for pollination may face population declines due to the decrease in pollinators [31,107]. Not coincidentally, excessive ALAN on tall buildings and structures will attract migratory birds, disorienting them and diverting them from their migration routes, causing them to miss ideal timing and conditions for nesting or feeding [32–34,108]. Worse, millions of birds die each year from impacts with bright-looking buildings [32,33,109]. Moreover, studies have shown that long-term exposure to excessive amounts of ALAN can have a detrimental or disruptive effect on the immune function and reproductive rhythms of some mammals, such as rodents [110] and marsupials [111], and even cause habitat erosion [112,113]. Studies from Australia, Israel, and the United States have shown that ALAN affects the egg-laying behavior of mother turtles on the beach and causes disorientation of hatchlings, preventing their return to the sea [35–37,114]. The glare from ALAN can reach the wetland habitats where amphibians such as frogs survive, disrupting their nocturnal calls and reproductive behavior [38,39,115]. Migrating fish are also confused by ALAN, leading to excessive energy loss and spatial barriers to migration [40,116,117]. Light pollution likewise distorts the circadian rhythms of plants, affecting their germination, flowering, dormancy, defoliation, and other phenological behaviors, causing changes in community structure, which in turn affects the balance of the ecosystem [41,42,118,119]. In summary, excessive ALAN exposure causes indisputable negative effects on the physiology of both animals and plants and even threatens the service functions of entire ecosystems [120–122]. Accordingly, ecological damage has now become an attractive subfield in light pollution research.

- **Energy waste.** Although the advancement of lighting technologies such as LEDs has reduced the energy consumption of individual light equipment, excessive ALAN leads to a large amount of electricity consumption, thus creating a general situation of inefficient lighting, triggering energy waste and environmental pressure, which is contrary to the policy of energy saving and carbon reduction [43,123]. The heat emitted by various devices and the operation of power supply facilities also causes a temperature rise, exacerbating the "Urban Heat Island Effect" and global warming. According to IDA, the U.S. consumes 120 TWh (i.e., 100 million kWh) of electricity per year to illuminate streets and parking lots, which is comparable to the two-year electricity consumption of the entire New York City [44]. Due to an illogical lighting scheme, about 30% of outdoor lighting in America is wasted, along with an annual loss of USD 3.3 billion and 21 million tons of $CO_2$ emissions, for which about 875 million trees must be planted per year to offset this detrimental consequence [124]. In brief, the energy wasted and the extra carbon emissions caused by light pollution are substantial, and the growth in carbon emissions further exacerbates the environmental burden, leading to a vicious circle.

- **Other impairments.** A significant portion of ALAN is emitted toward the sky above the city, substantially increasing the brightness of the natural sky background, limiting the ability of humans to capture stars with naked eyes [125]. The over-expansion of ALAN also affects astronomers' studies of the stars with the help of observational

instruments [20,126]. If there is an ALAN source within sight of the astronomical equipment, it will directly obscure the light from the target object, which greatly increases the exploration difficulty [127]. Another thing that most people may not be aware of is that today's world is gradually losing the beautiful dark sky together with its cultural values [128]. In that case, lots of young people growing up in cities may never be able to enjoy the spectacular scenery of the Milky Way galaxy in the future as the pure dark night fades away [47].

In general, according to the outcomes of content mining, the adverse effects brought by nighttime light pollution can be summarized as endangering human health, causing ecological damage, intensifying energy waste, hindering astronomical observation, and undermining cultural values, etc. Consequently, nighttime light pollution has gradually become one of the fastest deteriorating and most critical global environmental issues.

### 3.2.2. Monitoring Technologies of Nighttime Light Pollution

- **Monitoring based on field data.**

In field monitoring, specific instruments should be selected concretely to achieve specific requirements. Normally, a sky quality meter (SQM) is particularly designed to measure the brightness of stars and atmospheric transparency. The spectroradiometers are founded on the principle of spectrophotometry, which is competent in accurately measuring the tristimulus value, chromaticity, color difference, color rendering index, and illuminance [129,130]. The CCD cameras with fisheye lenses are invented to take full-sky images on the ground and obtain color signals, which can be further used for quantification of brightness after computer processing [131]. Additionally, under the control of radio equipment, unmanned aerial vehicle (UAV) carries digital cameras, RGB color sensors, SQM, etc., to obtain photography samples or parameter information [132,133]. According to the data attributes from the observations, the field measurement techniques for the urban nighttime light environment can be divided into two categories: image-based and value-based [134].

As for image-based methodology, most related studies were carried out in small areas, embarking on the improvement of hardware equipment and the measurement of visual metrics, and the raw measurements usually require post-processing with the aid of computers. In 2010, Kolláth, based on digital single reflex (DSLR), abstracted the artificial component of zenith brightness in the night sky as a function of distance from the city center and simulated the night sky light quality at the Zselic Starry Sky Park, one of the first "International Dark Sky Parks" in Europe, with the support of radiative transfer models [135]. In 2011, Falchi et al. first calibrated the color, brightness, and internal parameters of multiple CCD cameras to dynamically monitor the brightness of the sky affected by different levels of light pollution [136]. In 2017, Jin et al. took a two-dimensional color luminance meter equipped with a wide-angle lens to observe the ALAN at different periods, then analyzed the overall brightness, chromaticity, and color temperature distribution characteristics of civic spaces [137]. In 2019, Fiorentin et al. proposed an image-based approach (MINLU) using drones and balloons carrying components such as digital cameras and RGB color sensors in a confined space at relatively low altitude to measure the ALAN intensity of pollution sources and their spectral power density (SPD) with spatial resolution of ~0.1 m and temporal resolution of up to several hours [138]. In 2022, Massetti et al. took a rural residential area in Italy as their study area, loaded both digital cameras and SQM on a UAV for simultaneous exposures, and investigated the relationship between indices calculated from remotely sensed images and nighttime ground brightness (NGB) through regression models [139].

As for value-based methodology, most studies used the SQM as the primary data source, occasionally supplemented by other auxiliary data to monitor local nighttime light pollution. The SQM is a light environment measurement instrument designed by Unihedron, a Canadian company, with advantages such as automated measurement, fast referencing, portability, and high sensitivity [140]. In 2007, Kocifaj proposed a scalable

theoretical model of light pollution from terrestrial light sources based on SQM-based measurement values from optical instruments, which is applicable to calculate the angular distribution of diffuse light generated by various types of terrestrial light sources [141]. In 2012, Kuechly et al. used SQM to observe night sky brightness in Berlin, Germany, and analyzed the effect of cloud cover on urban nighttime light pollution [142]. In 2013, Ścieżor proposed a method combining archival observations and SQM to determine the brightness of stars in the cloud-free Polish night sky during 1994–2009, followed by a cross-validation of night sky brightness maps obtained from known models to quantify the local level of light pollution [143]. In 2014, Pun et al. used SQM to continuously measure the sky brightness over Hong Kong and found a severe nighttime light pollution problem [144]. In the same year, Puschnig et al. measured the night sky brightness at Potsdam by SQM and investigated the relationship between cloudiness and moon phase and night sky brightness by modeling analysis [145]. In 2021, Kolláth et al. used a Konica-Minolta CS2000A spectroradiometer to measure the spectral radiance of the night sky in Canada and proposed a new metric for the measurement of light pollution in the dark sky [146]. Compared to image-based methods, value-based methods are usually free of complicated post-processing and therefore have lower measurement errors, making them the dominant method for field measurement studies.

- **Monitoring based on remote-sensing data.**

In recent decades, with the launch of operational satellites and the innovation of remote-sensing information technology, a series of mature remote-sensing data sources for nighttime lights observation have been developed, including DMSP-OLS, NPP-VIIRS, EROS-B, LJ-1 01, Jilin1-03B, and Yangwang-1 ("Look up 1"). Among them, DMSP-OLS, NPP-VIIRS, and LJ-1 01 are nowadays widely adopted in monitoring nighttime light pollution by virtue of their larger volume, higher quality, and relative accessibility [147–153]. The basic properties of them are shown in Table 2.

**Table 2.** Main parameters of several ALAN remote-sensing data.

| Data | DMSP-OLS | NPP-VIIRS | LJ-1 01 |
|---|---|---|---|
| Waveband Range | 400–1100 nm | 505–890 nm | 460–980 nm |
| Data Type | 0–63 (digital number) | Absolute radiation value | Absolute radiation value |
| Satellite Launch Year | 1973 | 2012 | 2018 |
| Spatial Resolution | ~1000 m | ~500 m/~750 m | ~130 m |
| Temporal Resolution | 1 a | 1 a/1 month/1 d | 15 d |
| Available Time | 1992–2013 | April 2012–present | June 2018–February 2019 |
| Accessibility | Open-source | Open-source | Open-source |
| Quantified Digits | 6 bits | 14 bits | 15 bits |
| Data Source | NOAA | NASA/NOAA | Wuhan University |

In the 1970s, the U.S. Defense Meteorological Satellite Program (DMSP) carried a scanning radiometer (i.e., Operational Linescan System, OLS) with an amplitude of 3000 km, allowing for effective capture of very low radiation values [154]. As the first-generation satellite data for nighttime light published by NOAA, DMSP-OLS contains a stable light product (DMSPstl) that provides the most extended time series of images to date, which is popular in a range of studies related to urbanization and human activities [148,155–158]. In 2001, Cinzano et al. proposed a method to monitor nighttime brightness using DMSP-OLS imagery and generated the first global-scale map of night sky brightness, concluding that about two-thirds of the world's population lived in light-polluted areas [159]. In 2006, Chalkias et al. utilized DMSP-OLS images and GIS to model light pollution generation pro-

cesses in suburban areas, showing that suburbs were suffering from severe light pollution during urbanization [160]. In 2012, Cinzano et al. modified the light pollution transmission model proposed by Roy Garstang in 1986 using DMSP data and put forward an extended model (EGM) with the introduction of various factors, such as multiple scattering, Earth curvature, altitude, and atmospheric type, to provide a more generalized numerical method for quantifying the radiative transfer of light pollution in the atmosphere [161]. In 2017, Jiang et al. studied the light pollution characteristics of China at national, regional, and provincial scales using DMSP-OLS data with the linear regression method combined with the index method, revealing that the degree of light pollution in China was increasing, especially in the eastern coastal areas [162]. However, DMSP-OLS has several remarkable drawbacks, such as coarse spatial and temporal resolution, data oversaturation due to the low radiometric resolution of the sensor, the pixel blooming effect, and difficulty in time series comparison due to non-orbit correction [48,163]. Among them, the data oversaturation problem and the pixel blooming effect are the most prominent. The former is manifested in the sensor design defects lead to the light gray value of more than 63 areas of radiation information cannot be completely recorded, the latter is manifested in the sensor detects the night-light range is wider than the actual lighting range. The existence of these problems seriously reduces the quality of DMSP-OLS data, thus limiting its application scope and effectiveness. As a result, DMSP-OLS had been gradually replaced by other flourishing remote-sensing data in the past decade [82].

In 2011, NOAA-NASA launched a new generation of Earth observation satellite Suomi NPP, equipped with a Visible Infrared Imaging Radiometer Suite capable of acquiring the Day/Night Band (DNB), which can be employed for mapping global cloud-free nighttime imagery [150,164,165]. Compared with DMSP-OLS, NPP-VIIRS data are in-orbit corrected, with higher temporal and spatial resolution and radiometric quantization bits, do not suffer from oversaturation and the pixel blooming effect, and have cloud-contaminated pixels removed in the synthesis process. These multifaceted superiorities over previous generation data have made the NPP-VIIRS the most popular ALAN remote-sensing data, and some scholars have taken the night-light intensity of VIIRS products as a proxy for a true reflection of regional economic development [82,166,167]. The trend of literature publication after 2012 showed that light pollution studies based on NPP-VIIRS data had also proliferated (Section 3.1.2), resulting in an unprecedented expansion of the breadth and depth of research. In 2018, Hu et al. used NPP-VIIRS integrated annual images to investigate the potential association between nighttime light pollution and nesting density of three major sea turtles on Florida beaches and found that the nest density of all studied species showed a significant negative correlation with light pollution level [114]. In 2021, Yerli et al. investigated the spatial and temporal characteristics of light pollution in Turkey during 2012–2020 using VIIRS imagery in combination with the Astro GIS database and found a stable and continuous increase in ALAN in almost all Turkish cities [168]. In 2022, Bagheri et al. measured the sprawl dynamics and light pollution levels in Tehran and Tabriz, Iran, based on Landsat 7/8 images and VIIRS night-light images using BUNTUS, a fully automated algorithm with relative robustness to input data [169]. Nevertheless, limited by the relatively late launch of NPP satellites, only nearly a decade of data are available at present, which is still inadequate for decades-long time series studies. In addition, it is especially important to note that a prominent drawback of VIIRS data in practical applications is the sensor's lack of sensitivity to light in the 400 to 500 nm range (near-blue wavelengths), resulting in low observed radiance in that case [7,170]. When the VIIRS DNB data became available to the public, it was at the time when LED lighting was becoming common outdoors in many cities around the world [170,171]. Some Italian cities, such as Milan and Rome, have undergone comprehensive LED retrofitting of street lighting, which has led to a decrease in DNB radiance in the city center, but an increase in outlying areas [170,172]. However, due to the lack of long time series and higher-resolution multispectral noctilucent satellites nowadays, NPP-VIIRS data are still the mainstream of relevant analysis.

LJ-1 01, launched in 2018 under the auspices of Wuhan University, is the world's first satellite specifically focusing on ALAN. With a spatial resolution of about 130 m and excellent sensor sensitivity, LJ-1 01 is capable of collecting global data in 15 days under ideal conditions, and a single imaging process almost covers the spatial extent of a medium-sized city, which shows significant application potential [173]. However, due to the short service time of the satellite, the research results on light pollution evaluation are relatively few so far. Moreover, the spatial scope of the update of the high-resolution nighttime light data of Luojia-1 is still narrow, and global coverage has not yet been achieved, with more images available only in ALAN high-density areas such as North America, Europe, East Asia, and the Middle East [174]. In 2018, Jiang et al. analyzed the possibility of LJ-1 01 for monitoring nighttime light pollution and compared it with NPP-VIIRS data in various aspects. The results showed that LJ-1 01 has the same noise distribution pattern as NPP-VIIRS data but is more sensitive and scalable in detecting luminous intensity [175]. In 2022, Li et al. utilized LJ-1 01 data and developed various empirical models to evaluate the night environment in Nanjing, providing a quantitative reference for local lighting management [176,177]. The innovative attempts of the above studies for new data sources have, to a certain extent, promoted the research in the field of nighttime environment monitoring to a more sophisticated direction.

Photographs taken by astronauts on the International Space Station (ISS) are another referable source of remote-sensing data on nighttime light pollution. Compared with the single-color images taken on typical remote-sensing satellites, true-color photos obtained on the ISS, which is located in orbit about 400 km from the surface, provide richer color information on a large scale with a spatial resolution of 5–200 m [178]. These nighttime photos are usually taken with DSLRs, and they have been recorded from 2000 to the present [179]. Since this type of measurement is mostly controlled manually, it also has several obvious disadvantages compared to satellites located in specialized orbits, i.e., the sampling time and spatial distribution of the data are irregular and often do not provide global coverage. Moreover, the raw images require a series of corrective post-processing due to the skewed observation angle, which often results in observable errors [180]. However, ISS images are particularly serviceable in presenting light pollution-sensitive areas, and they contain affluent spectral information for environmental modeling of light pollution. For example, in November 2015, an astronaut aboard the ISS used DSLR to obtain a high-quality nighttime photograph of the city of Calgary, Canada, showing major waterways and main municipal parks in clear silhouette against surrounding artificial lights, which could effectively help light pollution researchers or dark sky protectors understand the distribution, color temperature, and SPD of ALAN within the city [178]. In 2019, Sánchez De Miguel et al. proposed a method to classify outdoor lighting types from ISS images and explored the relationship between spectral information and several key environmental indices (including melatonin suppression index, starlight index, photosynthesis index, carbon emissions, etc.) through a modeling approach in Milan, Italy [181].

### 3.3. Evolution of ALAN in Typical Countries from 2013 to 2021

Based on the remote-sensing data introduced in Section 2.1.2 and the analysis steps proposed in Section 2.2.4, the evolutions of the three NPP-VIIRS ALAN indicators of G20 countries over the period 2013–2021 are shown in Figure 6. In order to clearly manifest the fit of each trend line, this study presented the parameters of each linear fit in a list (Table 3). Moreover, the results of the Mann–Kendall test are presented in Table 4. The complete dataset for trend analysis and Mann–Kendall trend test procedure table can be found in the Supplementary Materials.

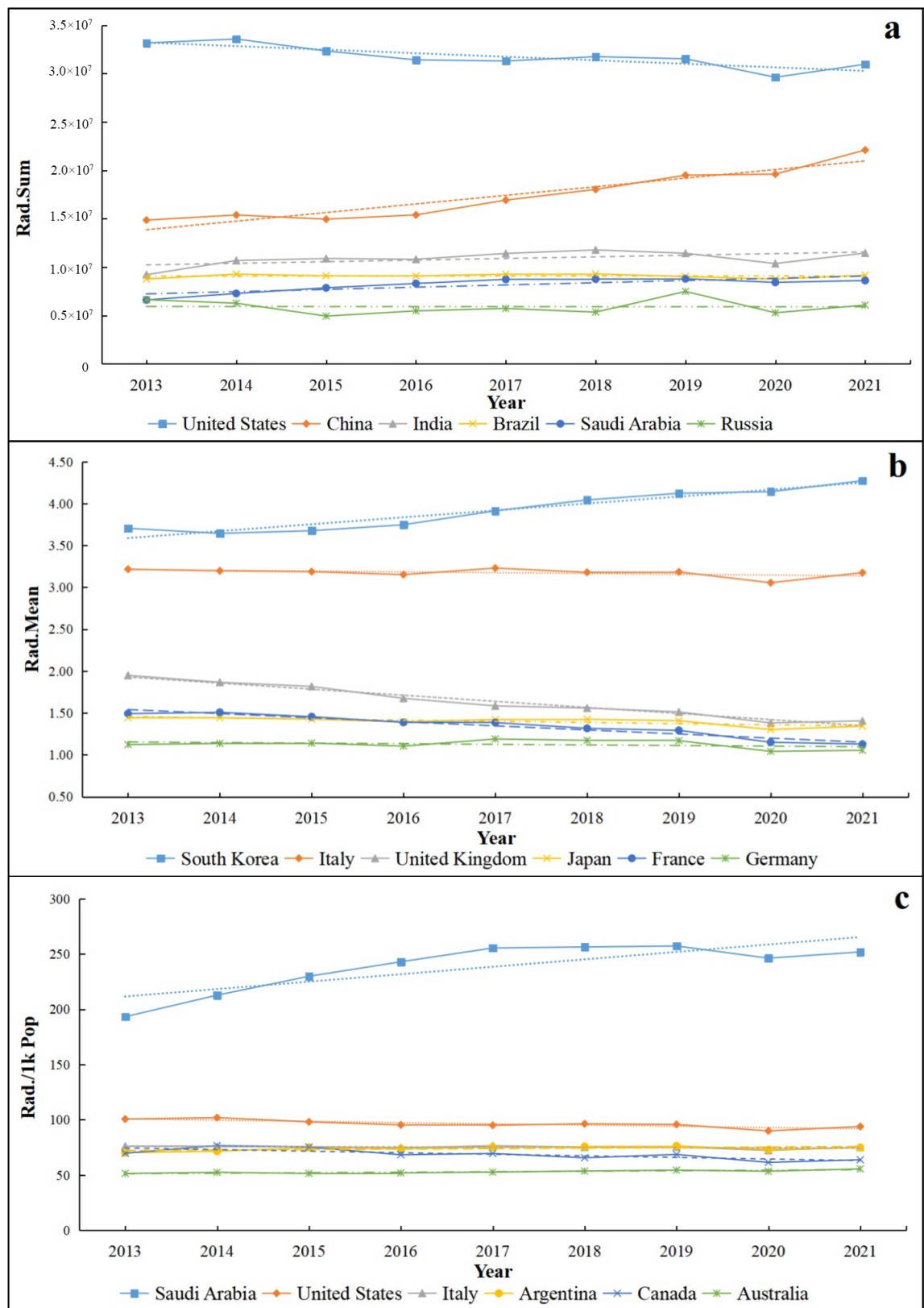

**Figure 6.** Evolutions of the three NPP-VIIRS ALAN indicators of G20 countries over the period 2013–2021. (Top six countries selected, respectively, and the dashed line indicates the linear fit trend). (**a**) Trend of total ALAN (***Rad. Sum***). (**b**) Trend of average ALAN (***Rad. Mean***). (**c**) Trend of ALAN per 1000 people (***Rad./1k Pop***).

**Table 3.** Results of the linear fit for the three indicators (corresponding to Figure 6).

| Indicator | Country | Slope | Intercept |
|---|---|---|---|
| | United States | $-3.65 \times 10^5$ | $3.36 \times 10^7$ |
| | China | $8.89 \times 10^5$ | $1.30 \times 10^7$ |
| *Rad. Sum* | India | $1.67 \times 10^5$ | $1.01 \times 10^7$ |
| | Brazil | $1.70 \times 10^3$ | $9.08 \times 10^6$ |
| | Saudi Arabia | $2.30 \times 10^5$ | $7.01 \times 10^6$ |
| | Russia | $-3.56 \times 10^3$ | $5.94 \times 10^6$ |
| | South Korea | $8.26 \times 10^{-2}$ | 3.51 |
| | Italy | $-9.48 \times 10^{-3}$ | 3.22 |
| *Rad. Mean* | United Kingdom | $-7.25 \times 10^{-2}$ | 2.00 |
| | Japan | $-1.40 \times 10^{-2}$ | 1.47 |
| | France | $-4.87 \times 10^{-2}$ | 1.59 |
| | Germany | $-7.13 \times 10^{-3}$ | 1.16 |
| | Saudi Arabia | 6.72 | 204.80 |
| | United States | $-1.11$ | 101.85 |
| *Rad./1k Pop* | Italy | $-0.22$ | 76.26 |
| | Argentina | 0.46 | 71.66 |
| | Canada | $-1.42$ | 75.86 |
| | Australia | 0.45 | 50.70 |

**Table 4.** Results of the Mann–Kendall trend test for the three indicators.

| Indicator | Country | Mann–Kendall Statistic (S) | Test Statistic (Z) | *p*-Value [2] | Sen's Slope (β) | Evolutionary Trend (of 90% Confidence) |
|---|---|---|---|---|---|---|
| | United States | $-24$ | $-2.3979$ ** [1] | 0.0165 | $-327,488.89$ | Decreasing |
| | China | 34 | 3.4405 *** | 0.0006 | 18,057.29 | Increasing |
| *Rad. Sum* | India | 18 | 1.7724 * | 0.0763 | 157,139.13 | Increasing |
| | Brazil | $-2$ | $-0.1043$ | 0.9170 | $-1733.55$ | No significant trend |
| | Saudi Arabia | 24 | 2.3979 ** | 0.0165 | 231,392.38 | Increasing |
| | Russia | $-4$ | $-0.3128$ | 0.7545 | $-41,028.50$ | No significant trend |
| | South Korea | 32 | 3.2320 *** | 0.0012 | 0.0864 | Increasing |
| | Italy | $-18$ | $-1.7724$ * | 0.0763 | $-0.0052$ | Decreasing |
| *Rad. Mean* | United Kingdom | $-34$ | $-3.4405$ *** | 0.0006 | $-0.0730$ | Decreasing |
| | Japan | $-26$ | $-2.6064$ *** | 0.0091 | $-0.0115$ | Decreasing |
| | France | $-34$ | $-3.4405$ *** | 0.0006 | $-0.0471$ | Decreasing |
| | Germany | $-4$ | $-0.3128$ | 0.7545 | $-0.0074$ | No significant trend |
| | Saudi Arabia | 24 | 2.3979 ** | 0.0165 | 6.7750 | Increasing |
| | United States | $-24$ | $-2.3979$ ** | 0.0165 | $-1.0214$ | Decreasing |
| *Rad./1k Pop* | Italy | $-17$ | $-1.6773$ * | 0.0935 | $-0.1146$ | Decreasing |
| | Argentina | 17 | 1.6773 * | 0.0935 | 0.5000 | Increasing |
| | Canada | $-24$ | $-2.3979$ ** | 0.0165 | $-1.5125$ | Decreasing |
| | Australia | 27 | 2.7255 *** | 0.0064 | 0.5000 | Increasing |

[1] *, **, and *** indicate passing significance tests with 90%, 95%, and 99% confidence levels, respectively. [2] The *p*-value indicates the significance test level and a smaller value indicates a higher probability of accepting the $H_1$ hypothesis, i.e., a more significant monotonic trend.

As shown in Figure 6 and Tables 3 and 4, for the period 2013–2021, for the indicator ***Rad. Sum***, in terms of the total value, the U.S. leads all countries in all years and is the largest contributor to total nighttime lighting, followed by China, India, Brazil, Saudi Arabia, and Russia. In terms of the trend of variation, according to the MK test results, the total ALAN of the United States shows a decreasing trend year by year, which is consistent with the linear fit results; the total ALAN of China, India, and Saudi Arabia shows an increasing trend, especially China, where the increasing trend of the total ALAN is the most remarkable; the total ALAN of Brazil and Russia does not show a significant evolutionary trend at the 90% confidence level. In terms of the rate of change, the United States has the maximum absolute value of β and shows the fastest rate of decline; Brazil has the minimum absolute value of β and shows the slowest rate of decline.

For the indicator ***Rad. Mean***, in terms of the total value, most of the countries with high average ALAN are developed countries with relatively small land areas, among which South Korea has the highest average ALAN, followed by Italy, U.K., Japan, France, and Germany. In terms of the trend of variation, according to the MK test results, only South Korea's average ALAN shows an increasing trend year by year, which is consistent with the linear fitting results; Italy, U.K., Japan, and France show a decreasing trend, especially U.K. and France, which have the most significant decreasing trend of their average ALAN; Germany's average ALAN does not show a significant evolution trend at the 90% confidence level. In terms of the rate of change, South Korea has the maximum absolute value of β and shows the fastest rate of increase; Italy has the smallest absolute value of β and shows the minimum rate of decrease.

For the indicator ***Rad./1k Pop***, in terms of the total value, Saudi Arabia has the highest per 1000 people ALAN exposure, leading the other countries in all years, followed by the United States, Italy, Argentina, Canada, and Australia. In terms of the trend of change, according to the MK test results, the ALAN per 1000 people in Saudi Arabia, Argentina, and Australia generally showed an increasing trend year by year, while the ALAN per 1000 people in the United States, Italy, and Canada generally showed a decreasing trend. In terms of the rate of change, Saudi Arabia has the maximum absolute value of β and shows the fastest rate of increase; Italy has the minimum absolute value of β and shows the slowest rate of decrease.

## 4. Discussion

### 4.1. A Joint Analysis of the Bibliometric Results and Night-Light Remote Sensing Data

Based on the national conditions of each country and combined with the results of the time series statistical analysis in Section 3.3, some differences and patterns between nighttime lighting and economic development or environmental protection in typical countries in the past decade can be summarized. On the one hand, the developed countries, represented by the United States and Canada, have a high level of urbanization and nighttime infrastructure construction, and ALAN has long reached saturation, which, together with the introduction of relevant policies and regulations, has gradually led to the decline of the mean ALAN. For example, the night sky protection and light pollution prevention bills and guidelines proposed by the U.S. states and other light environmental protection organizations, such as the Royal Astronomical Society of Canada, have made a more outstanding contribution to the optimization of light at night in various countries [182]. On the other hand, in the developing countries represented by China and India, whose priority is usually economic development, due to the rapid urbanization brought about by the expansion of city size and the construction of supporting facilities required for nighttime activities, ALAN continues to increase year by year and presents a potential state that will not be reduced in the short term [183]; even though some areas of these countries have begun to pay attention to nighttime light pollution, the speed of light pollution prevention and control has consistently failed to keep up with the pace of economic growth, leading to an increase in the total ALAN for a long time to come [184].

To further elucidate the intrinsic relationship between the global trends of nighttime light pollution and the response attitudes of countries, we plotted the time series of relevant literature publications during 2000–2022 based on the 15 countries involved in Figure 6 (5 more countries with the highest number of publications were also added to derive more objective conclusions). The 20 countries with a relatively high number of publications include representative developed countries from Europe, America, and Asia, such as the United States, the United Kingdom, Germany, and South Korea, as well as typical developing countries, such as China, India, Russia, and Brazil. The time series graph is shown in Figure 7. Through the comparative analysis of Figure 6 (characterizing the trend of nighttime light pollution) and Figure 7 (characterizing the scientific research investment and response actions of each country), the following viewpoints were derived.

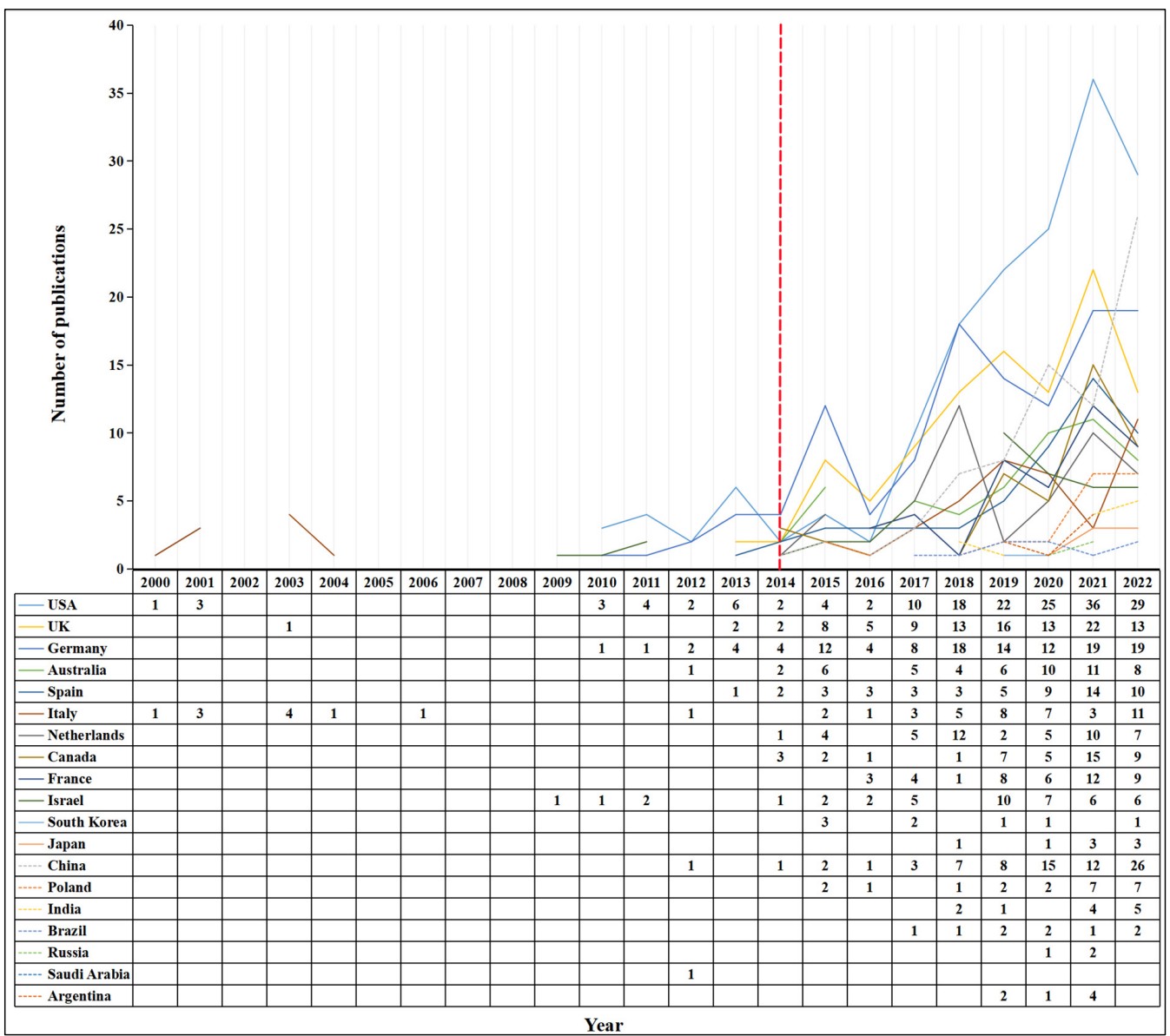

| | 2000 | 2001 | 2002 | 2003 | 2004 | 2005 | 2006 | 2007 | 2008 | 2009 | 2010 | 2011 | 2012 | 2013 | 2014 | 2015 | 2016 | 2017 | 2018 | 2019 | 2020 | 2021 | 2022 |
|---|---|---|---|---|---|---|---|---|---|---|---|---|---|---|---|---|---|---|---|---|---|---|---|
| USA | 1 | 3 | | | | | | | | | 3 | 4 | 2 | 6 | 2 | 4 | 2 | 10 | 18 | 22 | 25 | 36 | 29 |
| UK | | | | 1 | | | | | | | | | | 2 | 2 | 8 | 5 | 9 | 13 | 16 | 13 | 22 | 13 |
| Germany | | | | | | | | | | | 1 | 1 | 2 | 4 | 4 | 12 | 4 | 8 | 18 | 14 | 12 | 19 | 19 |
| Australia | | | | | | | | | | | | | 1 | | | 2 | 6 | 5 | 4 | 6 | 10 | 11 | 8 |
| Spain | | | | | | | | | | | | | | 1 | 2 | 3 | 3 | 3 | 3 | 5 | 9 | 14 | 10 |
| Italy | 1 | 3 | | 4 | 1 | | 1 | | | | | | 1 | | | 2 | 1 | 3 | 5 | 8 | 7 | 3 | 11 |
| Netherlands | | | | | | | | | | | | | | | 1 | 4 | | 5 | 12 | 2 | 5 | 10 | 7 |
| Canada | | | | | | | | | | | | | | | 3 | 2 | 1 | | 1 | 7 | 5 | 15 | 9 |
| France | | | | | | | | | | | | | | | | | 3 | 4 | 1 | 8 | 6 | 12 | 9 |
| Israel | | | | | | | | | | 1 | 1 | 2 | | | 1 | 2 | 2 | 5 | | 10 | 7 | 6 | 6 |
| South Korea | | | | | | | | | | | | | | | | 3 | | 2 | | 1 | 1 | | 1 |
| Japan | | | | | | | | | | | | | | | | | | | 1 | | 1 | 3 | 3 |
| China | | | | | | | | | | | | | 1 | | 1 | 2 | 1 | 3 | 7 | 8 | 15 | 12 | 26 |
| Poland | | | | | | | | | | | | | | | | 2 | 1 | | 1 | 2 | 2 | 7 | 7 |
| India | | | | | | | | | | | | | | | | | | | 2 | 1 | | 4 | 5 |
| Brazil | | | | | | | | | | | | | | | | | | 1 | 1 | 2 | 2 | 1 | 2 |
| Russia | | | | | | | | | | | | | | | | | | | | | 1 | 2 | |
| Saudi Arabia | | | | | | | | | | | | | | 1 | | | | | | | | | |
| Argentina | | | | | | | | | | | | | | | | | | | | | 2 | 1 | 4 |

**Year**

**Figure 7.** Time series of published literature for typical developed and developing countries (the red dashed line in the top half of the graph represents the time node when there was an abrupt change in the number of publications in developing countries. The solid lines in the legend in the bottom half represent developed countries, and the dashed lines represent developing countries). The complete data processing and mapping files can be found in the Supplementary Materials.

From Figure 7, it could be found that the research on nighttime light pollution in Europe and the United States (represented by the United States, Italy, and the United Kingdom) started earlier, and a few results were published in the early 20th century, which was the budding stage of scientific exploration. Research in developing countries generally began more than a decade later than in Europe and the United States, reflecting the disparity in the degree of importance attached to environmental pollution at different stages of economic development. As a country's economy continues to grow and national income continues to rise, its view of development generally progresses from "promoting the economy as the first priority" to "sustainable development with economic-environmental balance. After 2014 (marked by the red dashed line in the top half of Figure 7), it could be noted that there was a rapid leap in the number of publications not only in developed countries but also in developing countries, which indicates that light pollution had become a global environmental issue of widespread concern around the world.

A joint analysis of Figures 6 and 7 revealed that the number of publications and the *Rad. Sum* indicator (Figure 6a) were positively correlated. In general, countries with high nocturnal light intensity also produced more relevant research achievements. Among them, the U.S. had the first place in total nighttime light radiation, and the corresponding scientific research started earliest and yielded the most publications, showing great concern about nighttime light pollution for a long time. China ranked second in terms of total nighttime light radiation during this period and showed a quite fast growth rate, with a significant increase in related studies after 2014, indicating that the negative effects of light pollution brought about by high-speed economic growth have become increasingly prominent, and this environmental issue has received unprecedented attention. India, as a burgeoning developing country, ranked third in total radiation, but the relevant research started very late, and the scientific research results only showed rapid growth in the past two years. This lagging phenomenon may be more related to India's national conditions and development concept. However, with the continued urbanization and rapid expansion of the economy, it would be predicted that India's research efforts in the field of light pollution will grow further in the future.

Another interesting finding was that the light pollution levels in some developed countries were decreasing from the perspective of remote-sensing data (*Rad. Mean* indicator, Figure 6b), such as Italy and the U.K., where the number of publications was also relatively high in the early 21st century. This is because developed European countries such as Italy and the U.K. noticed the seriousness of light pollution early and adopted relevant prevention and control measures according to local reality. After various efforts such as legislation, monitoring, and prevention, the light pollution level was controlled within a reasonable range [185,186]. The law on lighting in public places at night enacted in Italy in 1985 set up standards for urban lighting in terms of brightness, height, and volume, on top of which different light standards were also set according to the age and color of buildings [187]. These measures not only enabled effective prevention of light pollution but also further improved the quality of the light environment. While South Korea is also a developed country, its *Rad. Mean* indicator was increasing year by year during 2013–2021, which was due to the fact that light pollution research in South Korea just started in 2015 (there was no relevant literature published before 2015), and the government authorities were slightly behind in paying attention to light pollution and taking corresponding measures, which had not yet formed a perfect and mature control system, resulting the level of light pollution was still in the rising stage.

In summary, the joint analysis of bibliometric results and remote-sensing data of nighttime light can intuitively reflect the impact level of nighttime light pollution and its variation trend in different countries under different development levels and governance concepts and also demonstrate the necessity of official intervention in light pollution control. The overall intensity and dynamics of light pollution, in turn, motivate countries to incorporate the sustainable concept of controlling light pollution levels to a reasonable

range into their national management systems when they are committed to economic growth.

*4.2. Comparison and Prospects of Light Pollution Monitoring Technologies*

As mentioned in Section 3.2.2, ground truthing using optical instruments characterized by numerous parameters is one of the main methods to quantify the urban nighttime environment. Since such techniques quantify physical parameters through contact or near observation, the measurements obtained are more consistent with the real conditions of the Earth's surface. Meanwhile, the time and location of ground measurements are usually controllable, which makes them more flexible, while the data measured can be cross-validated with the help of other abundant instruments. Nonetheless, in practical applications, the ground-based methods are prone to introducing errors caused by subjective conditions, and the measurement range is small and inefficient, making it difficult to adapt to the requirements of large-scale monitoring [167].

As the art, science, and technology of acquiring information about physical objects and the environment, remote sensing depends on non-contact sensors to record, measure, and interpret digital images [188,189], providing a fresh perspective on the Earth's observation. Traditional ground-based measurements are expensive, inefficient, small-scale, less informative, and challenging to compare with historical data because of the many-sided limitations of hardware [152]. In contrast, satellite-based observation exhibits a series of strengths such as wide observation range, outstanding information density, high spatial resolution, long time series, and integrability with other geospatial information technologies, which provides macro-scale details of the spatial distribution of light pollution [151,190,191]. Particularly, existing satellites designed to collect night-light data can efficiently generate regional and even global images of ALAN, bringing considerable convenience to estimating the urban light environment [9,149,156,192]. Unfortunately, the application of remote-sensing methods is limited by certain internal factors, such as the spatial resolution of the sensors and the scanned spectral bands, leading to problems such as light saturation effects and missing spatial details. Similarly, remote-sensing detection is also susceptible to external factors such as atmospheric conditions, cloud cover occlusions, and surface reflections, resulting in a slight lack of accuracy and fineness of data.

Given the advantages and disadvantages of both mainstream monitoring methods, there is an urgent need to combine the two means for a more comprehensive assessment of the nighttime environment. Currently, few studies focus on coupling remote sensing with ground-based measurements, mainly involving the comparison and calibration of measurements, the combination of hardware, or the integration of data. In terms of comparison and calibration of measurement results, in 2016, Katz and Levin discovered a robust correlation between ground truth data obtained by SQM and remote-sensing data represented by EROS-B, NPP-VIIRS, etc., and assessed the three-dimensional spatial nature of local light pollution in Jerusalem based on the calibrated results [191]. In terms of the combination of measurement hardware, several studies have tried to carry out more detailed aerial surveys of nighttime light pollution near the ground by aircraft or UAV with optical instruments (e.g., high-resolution cameras, SQM, etc.), and some scholars have carried out aerial observation experiments in typical cities and achieved satisfactory results [142,193]. In terms of the research on the fusion of the two data, because the ground-based measurement primarily observes the brightness, illuminance, and other parameters, while the remote-sensing observation principally provides DN value and radiation brightness, it is untoward to convert and calibrate between the two data, thus limiting the breakthrough progress of the research in this area, and there are still rare pertinent studies [163,194,195].

In the future, nighttime light pollution monitoring will evolve toward technological synergy (e.g., geographic information system and remote sensing, GIS, and RS) and integration of terrestrial and satellite data while continuously expanding the time scale and spatial scope [196]. In this regard, there is a pressing demand for all countries around the

world to work together to gradually build an integrated continuous ALAN observation network with ground-based instruments and satellite-based sensors as the front end and a GIS platform as the back end through technical research and information sharing. It is also necessary to enrich the visualization database of the nighttime environment and continuously promote new progress in its application in evolutionary trend research, spatial pattern exploration, influence factor modeling, and control decision making.

### 4.3. Management Recommendations for Protecting the Nighttime Environment

Urban nighttime light pollution management has become one of the most important subjects in the field of the ecological light environment in the world today, which requires a comprehensive approach from different perspectives, combined with multidisciplinary knowledge, to make it possible to effectively control the regional nighttime environment. As the results of the trend analysis in Section 3.3 show, the nighttime light pollution situation in each typical country is still an overall unpromising situation, and the oversupply of ALAN is still prominent and universal. With this in mind, this study proposed the following generic light pollution management recommendations for city managers to help manage the nighttime light environment more rationally by summarizing the advanced multi-country practical experiences.

#### 4.3.1. Improving Legal Norms

Improving planning and laws is the most effective way to prevent and control nighttime light pollution. Facing the increasingly tough situation of light pollution, many countries have started to pay attention to protecting the urban light environment in terms of legislation, which provides valuable references for other countries. As early as the 1980s, Japan enacted a single law on light pollution prevention and control, clearly defining light pollution and setting fitting rewards and penalties, creating a favorable atmosphere to protect the dark sky [197]. In 2003, the Czech Republic enacted a law on atmospheric protection, establishing administrative mechanisms for light pollution and encouraging autonomous regions to combat light pollution on the basis of local conditions while giving citizens the right to monitor and report light anomalies [198]. In 2007, Slovenia, which is known as the most beautiful stargazing place in the world, introduced its first light pollution prevention law in the form of a separate act, which strictly regulates the obligations of citizens to prevent light pollution, the use of shading equipment, the design standards of public streetlights, and the limitation of lighting hours [199].

In addition, the authorities are suggested to carry out environmental brightness zoning according to the nature and function of each space and set differentiated lighting standards to achieve fine control of light pollution [187,200,201]; as trees and bushes have been proven to block some light spilled in all directions, light pollution reduction through public landscaping improvements can be considered as well [202]. In order to make the remedies evidence based, it is also recommended to create a unified scientific evaluation standard for the light environment.

#### 4.3.2. Promoting Technology Innovation

Combining the prominent feature that the degree of light pollution is directly linked to the brightness of the light source, managers can improve the urban nighttime environment from the perspective of optimizing the light source itself. With the continuous advent of new technologies in public lighting, researchers and designers are increasingly concerned about the quality of the nighttime environment and are committed to designing energy-efficient lighting systems. For example, Garces-Jimenez et al. analyzed the ability of different artificial neural network (ANN) structures to simulate public lighting designs with different implicit layers to optimize multiple objectives and finally determine the most suitable lighting parameter configuration [203].

With the advancement of lighting technology and the artificial intelligence industry, there are already several mature, intelligent lighting products on the market that can

be introduced by city managers to effectively alleviate the phenomenon of illumination mismatch and energy consumption [204–207]. To quantify the effects of new intelligent lighting technologies on nocturnal species, in 2020, Bolliger et al. examined the number of insect and bat activities at night on streets with smart dimming streetlights and found that automated street dimming technologies helped mitigate the harmful effects of artificial lighting on these organisms at night [106].

Moreover, the authorities are also suggested to strengthen cooperation with research universities and professional institutions to manufacture lamps and lanterns, lighting arrangements, and landscape design to achieve coordination and unity to promote the construction of smart cities further.

### 4.3.3. Raising Awareness and Education

Surveys show that most people are not aware of the hazards of light pollution, and they do not even consider it a growing environmental problem [182]. Therefore, it is necessary to strengthen the publicity and education, increase the teaching content of "light pollution" in school classes, and make light pollution popularization through various media such as the Internet, cell phone, TV, and radio. In this way, the authorities will enhance the public's knowledge and understanding of light pollution, make them deeply aware of the hazards of light pollution, and motivate them to take measures within their ability to reduce the adverse influence, such as turning off lights before sleep (especially for infants), developing the habit of turning off lights if necessary, using curtains with the function of light blocking, and taking the initiative to report unreasonable outdoor lighting facilities, etc., to create a suitable light environment [21,208].

### 5. Conclusions

Rapid economic development has led to the over-expansion of outdoor artificial lighting in cities at night, resulting in an increasing phenomenon of light pollution worldwide, which has imperceptible negative impacts on human health, the ecological environment, energy utilization, and even astronomical observation that cannot be ignored. Without the support of scientific monitoring technologies and reasonable regulatory strategies, light pollution and its harmful effects caused by excessive nighttime lighting expansion will become even more catastrophic. On the basis of this, this study analyzes the current research status and trends of nighttime light pollution by systematically screening and condensing the relevant literature in the WOS database and summarizes the adverse effects of light pollution and monitoring techniques through keyword co-occurrence technology based on VOSviewer, and also explores the changing trends of nighttime light pollution in typical countries from 2013 to 2021 using remote-sensing data, and finally puts forward feasible management suggestions for protecting the nighttime environment. The results of the status analysis show that the research on nighttime light pollution has been rising in enthusiasm in recent years, and most of the countries with high contributions to the global economy or electricity consumption have a relatively high number of publications. In terms of negative impact, it can be found through the summary that the destructive effect of nighttime light pollution has penetrated various aspects, such as human health, the ecological environment, energy use, and even astronomical observation. In terms of measurement technology, the current mainstream approaches are divided into ground-based measurement and remote-sensing observation, which have their own advantages and disadvantages and need to be better combined in future studies for more efficient monitoring. In terms of management strategies, the differences in light pollution prevention and control can be reflected in the distribution of nighttime light pollution and its evolution trends in typical countries: most developed countries have a higher overall level of ALAN intensity, but the mean and per capita values are decreasing, which is attributed to the growing attention and stricter control of ALAN by their authorities; while developing countries' show the opposite trend, which is because the demands of economic development still outweigh environmental protection. In this regard, we call for greater cooperation among

countries around the world and suggest the regulators start from three aspects: legislation, technology, and education, to promote the construction of smart cities and sustainable economic development.

**Supplementary Materials:** The following supporting information can be downloaded at: https://www.mdpi.com/article/10.3390/rs15092305/s1. Figure S1: Per capita values of electricity consumption and GDP and number of pieces of literature published by country; Figure S2: Total value of electricity consumption, GDP and population, and number of pieces of literature published by country; Table S1: Research direction count table; Table S2: Table of the number of pieces of literature published in each country; Table S3: Visual word list based on VOSviewer; Complete dataset for trend analysis.xlsx; Mann–Kendall trend test.xlsx; List of 160 documents for intensive reading.xlsx; Time series of published literature in developed and developing countries.xlsx.

**Author Contributions:** Conceptualization, Y.Y. and C.H.; methodology, C.H.; data curation, Y.J. and B.L.; writing—original draft preparation, Y.Y. and C.H.; writing—review and editing, Y.Y.; visualization, Y.J. and B.L.; supervision, Y.Y. and Y.J.; project administration, Y.Y.; funding acquisition, Y.Y. All authors have read and agreed to the published version of the manuscript.

**Funding:** This research received no external funding.

**Data Availability Statement:** No new data were created for this study.

**Conflicts of Interest:** The authors declare no conflict of interest.

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
