# Peer review of "Research Progress, Hotspots, and Evolution of Nighttime Light Pollution: Analysis Based on WOS Database and Remote Sensing Data"

_remotesensing, doi:10.3390/rs15092305_

Round 1
Reviewer 1 Report
Your manuscript provides a very comprehensive overview of the current state of research in light pollution. I find it sufficiently informative for readers who will potentially do research on the topic. However, I have a number of questions regarding your manuscript:
1. While I understand that you used the VOSviewer as a blackbox program, it may help if you could explain a bit how it works (e.g. the working principle of its algorithm).
2. It is also understandable that you exclude literatures which are not written in English or Chinese, but this certainly will introduce positive bias towards publications of these two languages (which I believe is also why the US, China and UK have the highest numbers of publications). I cannot see a good workaround here, but it may perhaps help if you explicitly state this in Section 2, so that the readers are aware of this limitation.
3. You mentioned that you used the trend data from lightpollutionmap.info for your trend analyses. Did you use the EOG dataset, or did you use the newer and improved Black Marble dataset from NASA? In addition, please bear in mind that since VIIRS is not sensitive to blue light, a drop in radiance does not always mean that the amount of emitted light is decreased, especially in cases which the public lighting of a city/region is refitted by LEDs of higher color temperature (i.e. more blueish). For example, Some cities in Italy, like Milan, has undergone a remodeling of street lighting to LEDs, which decreases the radiance values on VIIRS–DNB, but the radiance has in fact increased (see as an example Fig. 5 of Kyba et al. 2017(DOI: 10.1126/sciadv.1701528)).
4. I wonder if you are able to produce a list of the 50 literatures for Figure 1. I believe that you have put at least some of them in the Bibliography section, but putting it into a separate list, probably as a supplemental document, will be nice.
Reviewer 2 Report
The authors aim to review the literature on night-time light pollution in terms of research progress, hot spots and developments. This study uses the keyword co-occurrence technique based on the WOS viewer to analyse the actuality of current research and trends on night-time light pollution through a retrospective review of the relevant literature and attempts to summarize the adverse effects and identify light pollution monitoring technologies . In addition, the trend of night time light pollution in typical countries from 2013 to 2021 is explored. The objectives, materials and data are well presented, and the map material illustrates the main results. The originality of this study is the integration of the specialized literature analysis method and remote sensing data analysis. Also, the article is well structured. However, it is not very clear why the authors integrated a literature review method and a remote sensing data analysis.
This study has a very good potential to be published after discussing some issues described below:
3.1.2. Section - title is inappropriate; this chapter analyses study frequency. A linear or (perhaps exponential) trend fit analysis is required to examine the trend.
3.2.2. The section should be moved to Methodology.
Table 3. Statistical testing of linear trend parameters should be added. I strongly suggest a non-parametric Mann-Kendall test.
1. A better explanation for choosing a single database (WOS) for this study. Why don't you also use the Scopus database?
2. A more detailed explanation in the Discussion section of the integration between a literature review method and a remote sensing data analysis is needed.
Reviewer 3 Report
The goals of the article are not clear. It presents itself as a study merging a comprehensive meta review of the field with remotely sensed data, but is mostly a review article with little to no original work.
The work is not detailed enough in its analysis of remotely sensed data and is mostly a review article.It would work fine as a review if the data analysis is removed, but if the goal is to show both the data analysis needs to be more thorough.
Section 2.1.2: A discussion of the advantages and limitations of using VIIRS as a reference are not discussed, specifically in relation to using it for data trends.
Section 2.2.1: This paragraph only contains a very methodical summary of what was done that is overly detailed but lacks justifications for the choices made. Why was the search limited to these specific terms? The low number of results also suggest that this research is overly restrictive.
Section 2.2.2: Similarly, the filtering criteria used are not described nor justified.
Section 2.2.4: Of the 3 metrics chosen, the usefulness of the second one, and to a certain extent the first one as well, isn't clear. Again, the descriptions are overly detailed (such as the odd definition of a sum as the product of the values and their frequencies) but lacks justification.
Section 3.1.1: This paragraph only describes the Figure, and there is no other reference to it in the text. It doesn't contribute anything to the article. It should either be analyzed or removed completely.
Section 3.1.3: The article mentioned that most publications come from the US, the UK and China, but only considered English and Chinese publications. Per capita values would be more interesting, and the evident bias should be discussed. Moreover, the correlation found between the publications, GDP and energy consumption could be due to the fact that these three variables are all correlated with population.
Figure 4: I would have like to see the per capita values as well, to emphasize the discrepancies in usage instead of population.
Figure 5: This Figure isn't clear enough. I would replace it with tabulated data.
Section 3.2.2: field data
The second sentence is way too long. Consider separating it.
Second paragraph, no mention is made of airborne sensors on planes, drones and balloons.
Third paragraph, other sensors should be mentioned as well.
Overall, this section if too short and doesn't properly account for all the measurement methods used.
Section 3.2.2: satellite data
The use of ISS images should be mentioned as well.
Limitations of the three satellites are barely mentioned.
This section is too short as well.
Section 3.3: I understand that only the first 6 countries are shown for readability purposes, but the full dataset should be provided, either as an annex or supplementary material.
The discussion presented is too short and doesn't account for a lot of factors. This section seems to be meant to only present data for later analysis, as such this text should probably be moved.
Table 3: Why the "Country/Region" label? Only countries are shown. The fitting equation should be replaced by two columns with the slope and offset to increase readability. R² is meaningless and should not be used in scientific works.
Section 4: Many discussions in this section are not well supported by the content of this article, and as such would benefit from better referencing
Reviewer 4 Report
Dear authors,
your paper has an asset as a review of publications and measurement methods. An important approach which is light pollution modeling is omitted in the paper, even if the title does not mention it.
Minor comments:
1, Section 2.1.2 - VIIRS is an instrument onboard the Suomi-NPP, but also onboard NOAA-20 and NOAA-21 satellites; there are three VIIRS units launched at the moment. Please, correct the statements if necessary.
2, Section 2.2.1 - How the TS is sensitive to the word combination? How do the results differ when one or more words are changed by e.g. skyglow, obtrusive light or ALAN?
3, Section 3.2.1 - There are notoriously known facts insection 3.2.1, but as a review with appropriate citations is fine. However, the readers maybe expect to summarize some new points of view, rather than well-known facts.
4, Line 344 - Please, add a reference to using SQM for measuring the brightness of stars.
5, Line 345 - Please, add a reference to a measurement of light pollution directly by spectroradiometers.
6, Line 347 - I expected the citations related to the specific type of instruments used for light pollution measurements. It should be useful for readers not familiar with a different types of unusual devices, e.g. UAV light intensity tester, range radar, etc.
7, Section 3.3 - What is the source data for these analyses?
8, Figure 6 - VIIRS data are noisy, even when aggregated over time. In the study, I did not find any information about the uncertainty uncertainties in the data, nor in Figure 6.
Round 2
Reviewer 1 Report
While I appreciate your effort in substantially editing the manuscript, I feel that the results of your analyses have now shown up some issues which may need to be addressed before publication:
Figure 3: the use of linear fit here is obviously inappropriate, so please consider using some higher-order fitting function, or simply omit that.
Figure 4: I cannot see any correlations between the presented quantities here. A scatterplot may be needed instead to show or disprove that there exists a relationship.
Figure 6: there are little evidence that the countries which you show a decreasing trend in radiance (e.g. Italy) did dim down in reality, due to the insensitivity of the VIIRS–DNB to blue light coincided with the increased use of blue-rich LEDs for public lighting (like in Rome, as mentioned in the first review report). Hence, this cannot support your arguments made in Section 4.1.
For suggested editings regarding the language, please check the attached annotations.
Last but not least, please state your source of funding for this study if possible.

Reviewer 3 Report
The authors have made significant changes and improvement to the article. The overall quality has been greatly improved.
Some minor corrections are still needed before publication, but mostly in the presentation of the content.
Figure 4: I don't see the relevance of the linear trendline. It should be removed.
Lines 554, 576, 581 and 589: Researchers should not be refered to by their first name only. At a minimum, last name should be used. Usage should also be uniform with other name referenced.
Lines 626, 628 and 640: "the Pixel Blooming Effect" does not need quotation marks, and I doubt the use of upper case is needed.
Line 664: represent -> such as
Line 687: remove "In addition"
